# Acetylcholine deficiency disrupts extratelencephalic projection neurons in the prefrontal cortex in a mouse model of Alzheimer's disease

Qingtao Sun [1,4], Jianping Zhang[2,4], Anan Li[1,3], Mei Yao[1], Guangcai Liu[1], Siqi Chen[1], Yue Luo[1], Zhi Wang[1], Hui Gong [1,3], Xiangning Li [1,3✉] & Qingming Luo [2✉]

Short-term memory deficits have been associated with prefrontal cortex (PFC) dysfunction in Alzheimer's disease (AD) and AD mouse models. Extratelencephalic projection (ET) neurons in the PFC play a key role in short-term working memory, but the mechanism between ET neuronal dysfunction in the PFC and short-term memory impairment in AD is not well understood. Here, using fiber photometry and optogenetics, we found reduced neural activity in the ET neurons in the medial prefrontal cortex (mPFC) of the 5×FAD mouse model led to object recognition memory (ORM) deficits. Activation of ET neurons in the mPFC of 5×FAD mice rescued ORM impairment, and inhibition of ET neurons in the mPFC of wild type mice impaired ORM expression. ET neurons in the mPFC that project to supramammillary nucleus were necessary for ORM expression. Viral tracing and in vivo recording revealed that mPFC ET neurons received fewer cholinergic inputs from the basal forebrain in 5×FAD mice. Furthermore, activation of cholinergic fibers in the mPFC rescued ORM deficits in 5×FAD mice, while acetylcholine deficiency reduced the response of ET neurons in the mPFC to familiar objects. Taken together, our results revealed a neural mechanism behind ORM impairment in 5×FAD mice.

[1] Britton Chance Center for Biomedical Photonics, Wuhan National Laboratory for Optoelectronics, MoE Key Laboratory for Biomedical Photonics, Huazhong University of Science and Technology, Wuhan 430074, China. [2] Key Laboratory of Biomedical Engineering of Hainan Province, School of Biomedical Engineering, Hainan University, Haikou 570228, China. [3] HUST-Suzhou Institute for Brainsmatics, JITRI, Suzhou 215123, China. [4] These authors contributed equally: Qingtao Sun, Jianping Zhang. ✉email: lixiangning@mail.hust.edu.cn; qluo@hainanu.edu.cn

Alzheimer's disease (AD) is a neurodegenerative disease characterized by memory loss and cognition impairment[1,2]. Short-term memory impairments are characteristic of AD and these impairments are recapitulated in AD mouse models as object recognition memory (ORM) deficits[3,4]. Short-term memory is impaired in sporadic AD, familial AD, and in asymptomatic carriers of familial AD with presenilin-1 mutation[5-9]. In human studies, the visual paired comparisons task is used to test ORM[5-9], which is comparable with the one-trial novel object recognition (NOR) test and delayed non-matching to sample in rodents[10,11]. A series of studies have shown that ORM impairment correlates with the prefrontal cortex (PFC) dysfunction in both AD patients and mouse models[12-16]. Morphological studies have shown decrease in spine density, dendritic complexity, and soma volume of the PFC pyramidal neurons in both AD patients and mouse models such as tau transgenic mice and 5×FAD mice[17-20]. Chronic two-photon imaging studies found impaired dendritic spine plasticity in layer V pyramidal neurons of the PFC of P301S tau transgenic mice and APP23 and APPswe/PS1deltaE9 (deltaE9) mice crossed with Thy1-GFP mice[21,22]. Recent whole brain imaging studies demonstrated that cortical layer V pyramidal neurons labeled in Thy1-GFP mice are mainly extratelencephalic projection (ET) neurons[23,24]. Functional studies indicate that ET neurons in the PFC play a key role in short-term memory such as working memory and ORM[16,25-27]. However, the circuit mechanism between dysfunction of ET neurons in the PFC and ORM impairment in AD is not well understood.

It has been reported that cholinergic neurons in the basal forebrain that innervate PFC degenerate in AD patients[28]. Several pharmacological studies have reported that administration of M1 cholinergic receptor agonist in the PFC of mice could effectively improve the cognition of the animals and their performance in object recognition test and water maze test[29-32]. Physiological studies have demonstrated that layer V pyramidal neurons in the PFC were specifically activated by acetylcholine via M1 cholinergic receptor[33,34]. Thus, the morphological, pharmacological, and physiological studies all indicated that the layer V ET neurons in the PFC and their connections with cholinergic inputs may be the key components for PFC to regulate cognitive functions. However, there were also some reports showing that overstimulation of M1 cholinergic receptors in the PFC impaired working memory and attention[35,36]. Therefore, it is necessary to perform further studies to confirm the role of ET neurons in the PFC and their circuits in the regulation of short-term memory such as ORM.

Transgenic animal models for AD have facilitated research on the pathogenesis of AD. Compared to other AD transgenic mice, the 5×FAD mouse model carries the most commonly found presenilin-1 mutation in familial AD patients and can rapidly develop severe amyloid pathology, neuron loss, and cognitive memory impairment[37]. This mouse model has been widely used in studying the circuitry and molecular alterations during AD pathogenesis[38-40]. In the present study, we employed viral tracing, optogenetics, chemogenetics, whole brain imaging and fiber photometry to investigate ET neurons in the mPFC in both wild type mice and 5×FAD mice[37]. We found that ET neurons showed decreased response to object recognition test in 5×FAD mice. Optogenetic manipulation of the somas and their axon terminals in the supramammillary nucleus (SUM) affected ORM expression. Both whole brain imaging of viral tracing and in vivo functional acetylcholine recordings with acetylcholine indicators revealed that mPFC ET neurons in 5×FAD mice received fewer cholinergic inputs from horizontal diagonal band, a brain region in the basal forebrain. The ORM impairment in 5×FAD mice could be rescued by manipulating the cholinergic fibers in the mPFC, while acetylcholine deficiency changed the response patterns of ET neurons in the mPFC to familiar objects. Our results reveal a circuit mechanism of how mPFC ET neurons and their input/output connections regulate ORM and provide potential treatment targets for cognition impairment in the AD patients.

## Results

**Object recognition memory impairment and deceased response of ET neurons in the mPFC of 5×FAD mice during novel object recognition test.** First, we set to examine the activity of ET neurons in the mPFC ($ET_{mPFC}$) of both wild type mice and 5×FAD mice with fiber photometry. We used Fezf2-CreER mice driver line[41] to target ET neurons. Fezf2 is a gene expressed in the deep layers of the cortex and mainly in ET neurons[42,43]. To validate the Cre expression specificity in Fezf2-CreER mice, we injected a Cre dependent AAV virus expressing green fluorescent protein (GFP) into the mPFC and induced the Cre recombination with tamoxifen three days after virus injection (Supplementary Fig. 1a). A month later, we embedded the virus infected mouse brain in resin and performed whole brain imaging with the fluorescence micro-optical sectioning tomography (fMOST) system[23] (Supplementary Fig. 1b). We found that GFP positive neurons strictly located in the deep layers of the mPFC, which is consistent with the expression patterns of Fezf2 in the neo-cortex, indicating the specificity of the animals we used. The GFP labeled fibers mainly distributed in the dorsal striatum, basal forebrain, limbic thalamus, lateral hypothalamic area, and mid-brain, while very few axons projected to the contralateral cortex (Supplementary Fig. 1c, d). These results demonstrated that Cre dependent virus in the mPFC of Fezf2-CreER mouse line mainly targeted ET neurons.

Next, we examined the behavior alterations in the 5×FAD mice. Previous studies have shown that 5×FAD mice exhibited impaired memory and cognition[37,44]. Specifically, there were reports showing that 5×FAD mice exhibit sex biased pathology progression and female animals showed more severe symptoms and Aβ (amyloid protein, Aβ) accumulation[45,46]. Therefore, we chose female 5×FAD animals in our following experiments. To examine the behavior impairment of 5×FAD mice, we assigned the 5×FAD mice to different behavior tests at different ages. In open field test, the traveling distance, mean velocity and time were similar between wild-type (C57BL/6 J) mice and 5×FAD (C57BL/6 J background) mice at the age of 2 months and 6 months, respectively (Supplementary Fig. 2a, b), which indicated that the locomotion ability of 5×FAD mice were not altered at these ages. However, in novel object recognition test (NOR), 6-month old 5×FAD mice spent less time with novel objects compared to C57BL/6 J mice (Supplementary Fig. 2c-e), which indicated that 6-month-old 5×FAD mice exhibited impaired ORM. These results were consistent with a previous study[37]. Therefore, in the following experiments, we chose to use 6-month old 5×FAD mice. Then, we tested the effect of different delay times in the NOR according to the previous studies[47]. Although the animals showed worse performance with longer delay time (Supplementary Fig. 2e), the differences were not significant. We also explored the correlation between the object recognition index and the Aβ density in the mPFC (Supplementary Fig. 2f, g). The results showed that the object recognition index and the Aβ density showed very weak negative correlation, indicating that the heavier Aβ burden in the mPFC may result in worse behavior performance. However, there may be other factors that affect the behavior performance of the animals, since the overexpression of APP in the animal model may have many other impacts apart from Aβ accumulation.

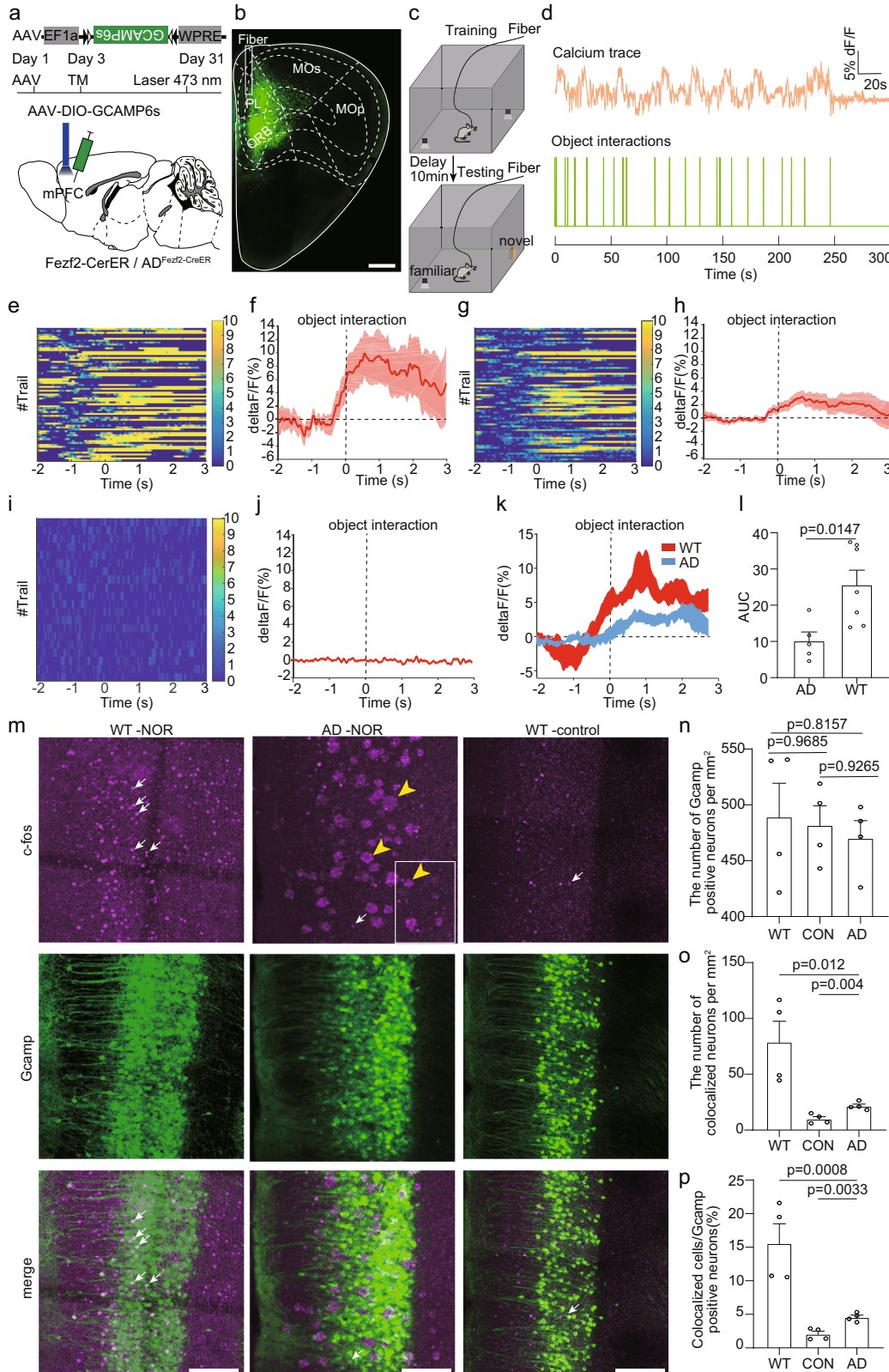

To monitor neuronal activities, a Cre dependent AAV virus expressing GCaMP6s was injected into the mPFC of Fezf2-CreER mice or $AD_{Fezf2\text{-}CreER}$ mice (5×FAD mice inbred with Fezf2-CreER mice) at five months old and we induced Cre recombination with tamoxifen three days after virus injection (Fig. 1a, b). To explore whether the cross between 5×FAD animals and Fezf2-CreER animals has any effect on the Cre expression, we conducted fluorescent in situ hybridization against Cre to examine the Cre expression in the mPFC of both Fezf2-CreER mice and $AD_{Fezf2\text{-}CreER}$ mice (Supplementary Fig. 3a–d). We found that the number of Cre positive neurons in both groups showed no differences (Supplementary Fig. 3e), indicating that the cross between 5×FAD animals and Fezf2-CreER animals did not affect the number of Cre+ neurons in the mPFC. One month

**Fig. 1 Altered response of extratelencephalic projection neurons to objects in the mPFC of 5×FAD mice. a** The experimental strategy of recording the activity of extratelencephalic projection neurons in the mPFC with fiber photometry. **b** An example image showed the virus injection site and the position of the optical fiber. **c** The experimental procedure of novel object recognition test. **d** An example showed the trace of the calcium signal and the time point of the object interaction during object recognition test. **e** Heatmap of $Ca^{2+}$ transients of extratelencephalic projection neurons in PL of Fezf2-CreER mice during object interactions. **f** Average plots across different trials of individual animals show that object interactions activated extratelencephalic projection neurons in mPFC. **g** Heatmap of $Ca^{2+}$ transients of extra telencephalic projection neurons in PL of AD$_{Fezf2-CreER}$ mice during object interactions. **h** Average plots across different trials of individual animals show that object interactions activated extratelencephalic projection neurons in mPFC of 5×FAD mice. **i** Heatmap of $Ca^{2+}$ transients of extratelencephalic projection neurons in PL of Fezf2-CreER mice during object interactions. The GFP was expressed in the PL instead of GCaMP6s. **j** Average plots across different trials of individual animals show that object interactions did not evoke clear response in GFP positive neurons. **k** The average plots of the calcium response from different animals. **l**. The quantification of area under curve (AUC) in **k**. two-tailed unpaired *t* test was used, Fezf2-CreER mice, $n = 7$ animals; AD$_{Fezf2-CreER}$, $n = 5$ animals. **m** Immunostaining against c-fos to validate the extratelencephalic projection neurons were activated in the object recognition test. White arrow indicates c-fos, yellow arrow indicates Aβ plaques. **n** Quantification of the number of extratelencephalic projection neurons that expressed GCaMP in the mPFC. $n = 4$ animals. One-way repeated-measures (RM) ANOVA with Tukey's post hoc test. **o** Quantification of number of c-fos positive extratelencephalic projection neurons in the mPFC. one-way repeated-measures (RM) ANOVA with Tukey's post hoc test, $n = 4$ animals. **p** Quantification of the percentage of c-fos positive extratelencephalic projection neurons in the mPFC. one-way repeated-measures (RM) ANOVA with Tukey's post hoc test, $n = 4$ animals. Scale bar in b is 500 μm. Scale bar in m is 100 μm. All data are listed as the Mean ± SEM. The color bars in the figure indicated df/f (%). NOR, novel object recognition test. TM, tamoxifen.

later, we assigned the test mice to the NOR and performed the fiber photometry as previous studies[48,49] (Fig. 1c). During object recognition test, interaction with objects (novel and familiar objects combined together) could evoke ET neuronal activities in both Fezf2-CreER mice and AD$_{Fezf2-CreER}$ mice (Fig. 1d–h). To confirm that the calcium signal fluctuation was associated with the interaction rather than caused by random locomotion of the test animal, we also performed fiber photometry on Fezf2-CreER mice that expressed GFP instead of GCaMPs in the mPFC. We found that fiber photometry recorded very little fluctuation in mice that expressed GFP (Fig. 1i, j), validating that the calcium signals were associated with the object interactions. Furthermore, we found that the amplitude of calcium signals in Fezf2-CreER mice was stronger than that in AD$_{Fezf2-CreER}$ mice (Fig. 1k, l, Supplementary Fig. 4a, b), indicating that the response during NOR was decreased in ET neurons in AD$_{Fezf2-CreER}$ mice compared to that in Fezf2-CreER mice. We also compared the total number of calcium transients, number of object interaction times, and velocity between Fezf2-CreER mice or AD$_{Fezf2-CreER}$ mice, and they showed no difference between the two groups (Supplementary Fig. 4c, d and g). The number of calcium transients showed strong correlation with the number of object interaction times but not with the velocity of the animals (Supplementary Fig. 4e, f, h and i). The amplitude of calcium signals was also not correlated with the velocity of the animals (Supplementary Fig. 4j, k). To further confirm that the response of ET neurons in the mPFC of 5×FAD mice was decreased compared to the wild type mice, we divided the Fezf2-CreER mice and AD$_{Fezf2-CreER}$ mice used for fiber photometry into an experimental group and a control group. In the experimental group, the mice were assigned to NOR. In the control group, the mice were kept in their home cages. One and half hours after the test, both groups of mice were sacrificed, and the brain slices containing mPFC were collected for c-fos staining (Fig. 1m). The number of GCaMP positive neurons between different groups showed no obvious difference (Fig. 1n). However, the number of c-fos and GCaMP double positive neurons in the mPFC of mice in the experimental group was larger than that in the control group (Fig. 1o), indicating that NOR did activate these ET neurons. Moreover, within the experimental group, compared to the AD$_{Fezf2-CreER}$ mice, there were more c-fos and GCaMP double positive neurons in the mPFC of Fezf2-CreER mice (Fig. 1o, p), which validated that the response of ET neurons of Fezf2-CreER mice was stronger than that of AD$_{Fezf2-CreER}$ mice during NOR.

We also examined the alterations of c-fos expression in the downstream subcortical regions of the mPFC. We found that after

NOR, c-fos neurons appeared in a lot of downstream subcortical regions of the mPFC, such as anterior olfactory nucleus, horizontal diagonal band, substantia innominata (SI), paraventricular thalamic nucleus, lateral hypothalamic area, periaqueductal gray, SUM, ventral tegmental area (VTA) and substantia nigra, reticular part (Supplementary Fig. 5a–k). Compared between the experimental group and the control group, the number of c-fos positive neurons significantly increased in the anterior olfactory nucleus, SI, SUM, VTA and substantia nigra, reticular part (Supplementary Fig. 5l). These results indicated that NOR activated the neurons in these brain regions. Within the experimental group, compared between the Fezf2-CreER group and the AD$_{Fezf2-CreER}$ group, the number of c-fos positive neurons was smaller in SI, SUM and VTA of 5×FAD experimental group (WT vs 5×FAD in SUM, $p = 0.11$, Supplementary Fig. 5l). These results indicated that AD development affected the neuronal response to NOR in these brain regions. To assure that the c-fos expression decrease in the mPFC, SI, SUM and VTA were due to the decreased neuronal activity rather than the neuronal loss in these brain areas, we also checked the number of neurons in these brain areas and compared the differences between wild type (C57BL/6 J) animals and 5×FAD mice. The immunohistochemical staining results against NeuN showed that at six months of age, the number of NeuN positive neurons in these brain areas showed no differences (Supplementary Fig. 6), indicating that in 5×FAD mice, there is no neuron loss in these brain areas at six months of age.

**Manipulation of activities of ET neurons in the mPFC affects object recognition memory expression.** We have already found that the response of ET neurons in the mPFC of 5×FAD mice was smaller than that in the wild type mice (Fig. 1). Therefore, we would like to figure out whether activation of these neurons could rescue the ORM impairment in 5×FAD mice. First, we expressed ChR2 in the mPFC of AD$_{Fezf2-CreER}$ mice at five months old with a Cre dependent AAV virus and induced Cre recombination with tamoxifen three days after virus injection (Fig. 2a, b). One month later, we assigned the test mice to NOR and performed optogenetic experiments by activating the somas of these mPFC neurons with 473 nm laser (Fig. 2c, 10 Hz, 15 ms pulse width, 3.5 mW at the tips of the optical fiber). Compared to the tests without light activation, light activation in 5×FAD mice significantly increased the novel object exploration time (Fig. 2d–f), indicating that the activation of ET neurons in the mPFC of 5×FAD mice improved the ORM expression. However, activation of ET neurons also significantly increased locomotion of 5×FAD mice (Fig. 2f). We also performed the light stimulation experiments in the control

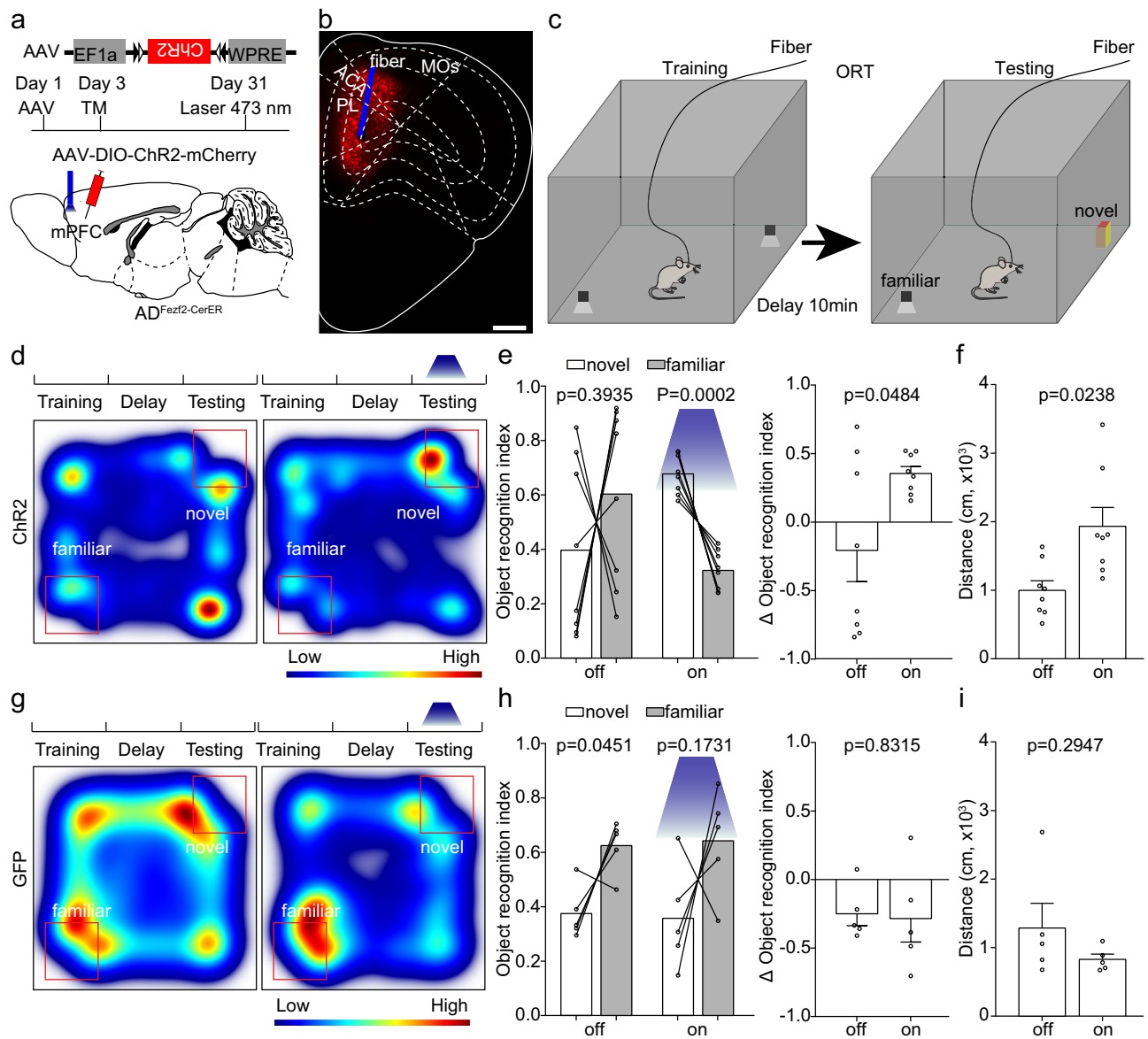

**Fig. 2 Activation of extratelencephalic projection neurons in the mPFC of 5×FAD mice improved the object recognition memory expression. a** The experimental strategy of activation of extratelencephalic projection neurons in the mPFC. **b** An example of position of optical fiber for activation of the extratelencephalic projection neurons in the mPFC. **c** The experimental procedure of novel object recognition test. **d** Heat-map plots showed object recognition memory measures from the object recognition test. Red = more time, blue = less time. The heatmap in the left and right showed examples of AD_Fezf2-CreER mice that expressed ChR2 in the ET neurons in the mPFC exploring the familiar object and novel object during the test session without or with light stimulation, respectively. **e** Statistical plots of object recognition index and delta object recognition index in laser off and laser on sessions. (left and right, two-tailed paired *t* test). *n* = 8 animals for each group. **f** Statistical plots showed the traveling distance during the test session with or without light stimulation. two-tailed paired *t* test; *n* = 8 animals. **g** The heatmap in the left and right showed examples of AD_Fezf2-CreER mice that expressed GFP in the ET neurons in the mPFC exploring the familiar object and novel object during the test session without or with light stimulation, respectively. **h** Statistical plots of object recognition index and delta object recognition index in laser off and laser on sessions. (left and right, two-tailed paired *t* test). *n* = 5 animals. **i** Statistical plot showed that the light stimulation in control group that expressed GFP did not alter the traveling distance during the test. two-tailed paired *t* test, *n* = 5 animals. Scale bar in b is 500 μm. All data are listed as the Mean ± SEM. ACA anterior cingulate area, PL prelimbic area, MOs secondary motor area, TM tamoxifen.

group in which the test animals expressed GFP instead of ChR2. We found that in the control group, light stimulation did not improve the preference of the animals to novel object or locomotion of the mice (Fig. 2g–i). These results demonstrated that light stimulation alone is insufficient to alter the exploration behaviors of the animals.

Next, we intended to explore how the manipulation of ET neurons in the mPFC of Fezf2-CreER mice affected ORM. We expressed inhibitory light sensitive ion channel NpHR into the

Fezf2-positive neurons in the mPFC with AAV virus (Supplementary Fig. 7a, b). One month after the virus injection, we assigned the test mice to NOR and suppressed the activity of Fezf2-positive neurons in the mPFC with 570 nm laser (Supplementary Fig. 7c, insistent inhibition, 3.5 mW at the tip of the optical fiber). Since the NOR contained three phases: training (memory formation), delay (consolidation), and testing (memory expression). We tested the effects of inhibition of ET neurons in the mPFC in these three phases separately. Interestingly, we

found that inhibition of ET neurons in the training phase and delay phase did not affect ORM (Supplementary Fig. 7c, d). Only during the test phase, inhibition of ET neurons in the mPFC disrupted the ORM expression (Supplementary Fig. 7e). These results indicated that the activity of these mPFC neurons is necessary for ORM expression but not for formation and consolidation. The inhibition of Fezf2-positive neurons in the mPFC also did not affect the locomotion of the animals during the test (Supplementary Fig. 7f).

Furthermore, to examine whether activation of ET neurons in the mPFC of Fezf2-CreER mice can further enhance the ORM, we injected a Cre dependent AAV virus expressing channelrhodopsin-2(ChR2) into the mPFC of Fezf2-CreER mice (Supplementary Fig. 8a, b) and tested the mice for NOR (Supplementary Fig. 8c). Conversely, we found that activation of these neurons during the training and delay phase did not affect ORM (Supplementary Fig. 8c, d). Surprisingly, activation of them in Fezf2-CreER mice disrupted the expression of ORM (Supplementary Fig. 8e). These results indicated that the ET neurons in the mPFC did not bidirectionally regulate the expression of object memory. The hyperexcitability of the ET neurons in the mPFC can also impair the ORM. Activation of the ET neurons in the mPFC of Fezf2-CreER mice also increased the locomotion of the animals (Supplementary Fig. 8f), which is consistent with our manipulation in the $AD_{Fezf2-CreER}$ animals.

Previous studies have reported that not only mPFC, but the hippocampus is also important for memory expression[50]. However, there were reports indicating that the ventral hippocampus is not crucial for object memory expression[50,51]. To test this implication and to explore whether activation of ventral hippocampus can rescue the ORM impairment in 5×FAD mice, we expressed ChR2 in the ventral hippocampus of the 5×FAD mice. One month later, we assigned the animals to NOR. Activation of the ventral hippocampus (10 Hz, 15 ms pulse width, 3.5 mW at the tips of the optical fiber) did not improve the exploration time towards novel objects or the locomotion of the animals (Supplementary Fig. 9), indicating that activation of the ventral hippocampus cannot rescue the ORM impairment in 5×FAD mice.

**ET neurons in the mPFC projecting to SUM regulated object recognition memory**. Previous studies have shown that the different group of ET neurons in the mPFC could separately encode locomotion and other behaviors[52–54]. Therefore, it is possible that there are different group of neurons responsible for encoding locomotion and ORM, respectively. In previous results, we have already identified the neuronal activities of several mPFC downstream areas were also affected by AD (Supplementary Fig. 5l). Therefore, we would like to investigate whether activation of the axon terminals of mPFC ET neurons in these brain regions could improve ORM. To identify the specific pathway responsible for the expression of ORM, we repeated the NOR and performed the optogenetics by manipulating the axon terminals of these mPFC neurons in different downstream brain areas. We found that activation of the axon terminals of mPFC in the SUM, instead of the SI, with 473 nm laser significantly improved ORM of 5×FAD mice (Fig. 3a–c, 10 Hz, 15 ms pulse width, 3.5 mW at the fiber tips). While activation of the axon terminals in the VTA significantly improved the traveling distance but not the object recognition ability during the test (Fig. 3c, e). In the control group, light stimulation of the axon terminals that expressed GFP did not alter the preference of the animals to novel object or locomotion (Fig. 3d, f). These results were consistent with a previous study that the mPFC-VTA pathway is responsible for encoding

locomotion[54]. Activation of the axon terminals in the SI did not affect object recognition or locomotion (Fig. 3c, e). Conversely, we found that inhibition of the axon terminals in the SUM disrupted the ORM of Fezf2-CreER mice (Supplementary Fig. 10). Inhibition of the axon terminals in the VTA decreased the traveling distance during the test but did not affect the preference of the animals to novel object. Inhibition of the axon terminals in the SI did not affect object recognition or locomotion (Supplementary Fig. 10). These results demonstrated that specific group of neurons in the mPFC regulated object recognition and locomotion separately.

**Anatomical differences between mPFC neurons projecting to SUM and SI**. Since manipulation of the axon terminals in the SUM and SI have different impacts on ORM, we would like to investigate whether these terminals were from the same neurons or different neurons. We injected two retrograde tracers (cholera toxin B subunit, CTb) with different fluorescence into SI and SUM, respectively (Fig. 4a–c). Seven days later, we found that different neurons in the mPFC were labeled by different CTb tracers (Fig. 4d), which showed distinct distribution patterns. The SUM projection neurons mainly located at the layer Vb of the mPFC, while the SI projection neurons mainly located at the layer Va and layer VI (Fig. 4d, e). The quantitative result showed that the overlap between the two groups of neurons was low (Fig. 4f). These results indicated that SUM projection neurons and SI projection neurons in the mPFC mainly were different neurons. To further characterize the morphological differences between the two group of neurons in the mPFC, taking advantage of the three-dimensional whole-brain imaging technique, we performed sparse labeling and precise whole-brain imaging to reconstruct the full morphology of the SUM projection neurons and SI projection neurons in the mPFC. Briefly speaking, we injected the CAV virus expressing Cre-dependent Flp recombinase (CAV-FLEx-Flp) into either SUM or SI of Fezf2-CreER mice. Meanwhile, Flp-dependent AAV virus expressing GFP was injected into the mPFC (Fig. 4g). We induced the expression of Cre with tamoxifen three days after the virus injection. One month later, we found that there were GFP labeled neurons in the mPFC which projected to SUM or SI, respectively (Fig. 4h). We embedded these brain samples in resin and performed whole-brain imaging with fMOST with a resolution of $0.32×0.32×2 \ \mu m^3$. Then, we reconstructed the neuron morphology and registered the data into the Allen common coordinate framework (Fig. 4j and Supplementary Fig. 11). SUM and SI projection neurons in the mPFC exhibited different axonal patterns: the SUM projection neurons mainly projected to the hypothalamus and midbrain such as lateral hypothalamic area, dorsomedial hypothalamic nucleus, VTA and substantia nigra, reticular part (Fig. 4j and Supplementary Fig. 11). These neurons were defined as the pyramidal tract projection neurons (PT neuron). The SI projection neurons consisted of two different projection patterns. One group of SI projection neurons mainly projected to orbital cortex, anterior cingulate cortex, anterior olfactory nucleus, nucleus accumbens (ACB), and dorsal striatum (CP) (Fig. 4j and Supplementary Fig. 11). These neurons were defined as the intratelencephalic projection neurons (IT neuron). The other group of SI projection neurons mainly projected to limbic thalamic nuclei (Fig. 4j and Supplementary Fig. 11). These neurons were defined as corticothalamic projection neurons (CT neuron). To further reveal the differences among three types of neurons, we quantified the dendritic lengths of these three types of neurons. We found that compared to the IT neuron, the CT and PT neurons had longer dendritic lengths (Fig. 4i), which was consistent with our previous study[55].

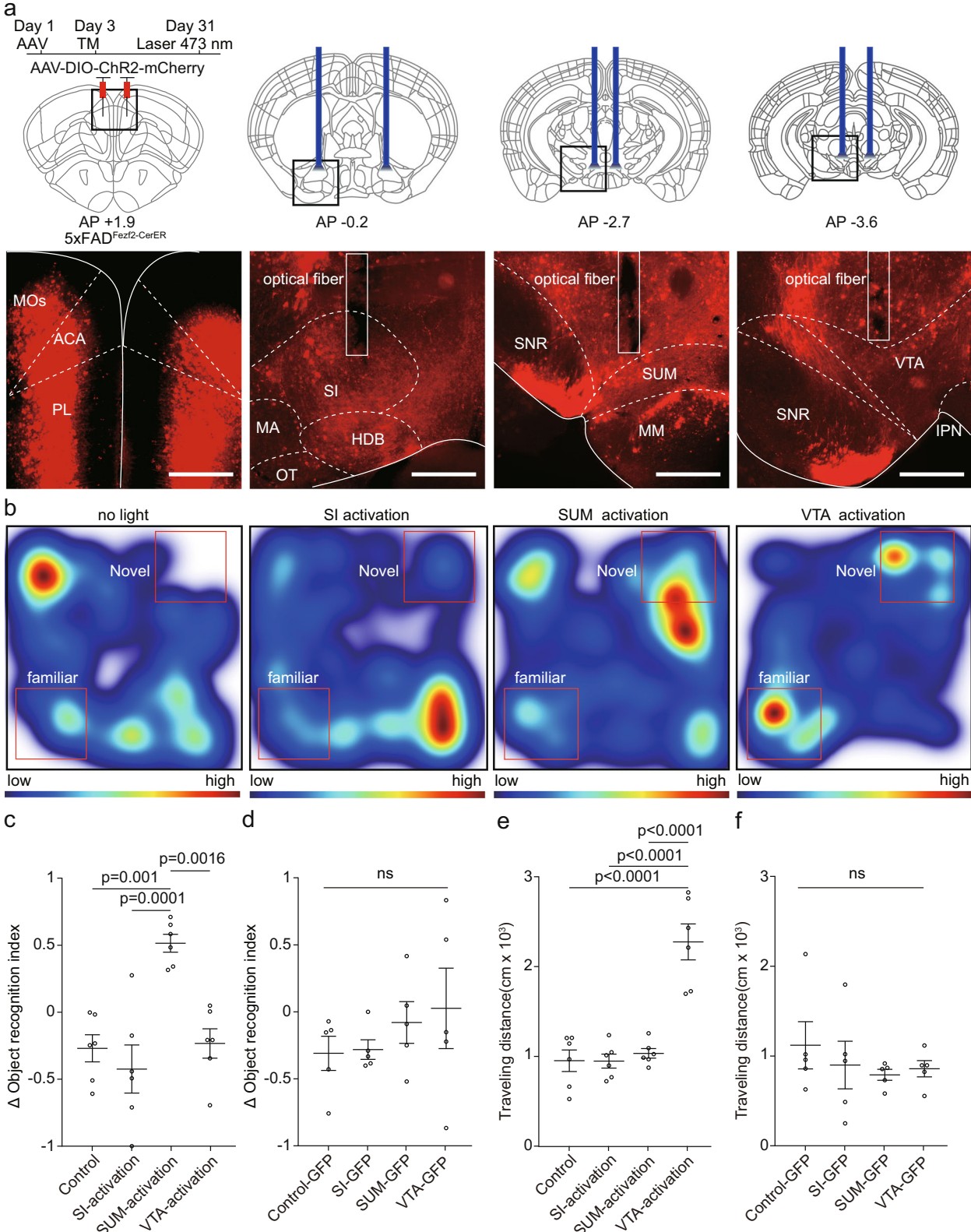

**Long-range inputs to ET neurons in the mPFC**. Previous studies have shown the decreased spine density, dendritic complexity, and soma volume in the deep layer pyramidal neurons of the PFC in both AD patients and 5×FAD mice[17–20]. The dendritic spines are the postsynaptic components of excitatory synapses, and the decreased spine density in the deep layer pyramidal neurons of the mPFC in 5×FAD mice indicated that these neurons received fewer excitatory inputs, which probably is the main cause of decreased activities. To find out which brain areas sent fewer inputs to ET neurons in the mPFC in 5×FAD mice, we employed whole-brain screening for the input neuron with rabies virus as in a previous study[56]. In our previous behavior screening, we have identified that the 5×FAD mice showed ORM impairment on an age-dependent manner (Supplementary Fig. 2). We assumed that

**Fig. 3 Activation of specific axon terminals of extratelencephalic projection neurons in the mPFC of 5×FAD mice improved the object recognition memory expression. a** The experimental strategy of terminal activation of extratelencephalic projection neurons in the mPFC of 5×FAD mice. The optical fibers were implanted in SI, SUM, or VTA bilaterally to activate the axon terminals in these brain areas (top). The virus injection site and the positions of the optical fiber were shown below. **b** Heat-map plots showed object recognition memory measures from the object recognition test under different experimental conditions (no light stimulation; activation of axon terminals in SI; activation of axon terminals in SUM; activation of axon terminals in VTA.) Red = more time, blue = less time. **c** Statistical plots showed the delta object recognition index under different conditions in experimental group (mice that expressed ChR2 in the mPFC). one-way repeated-measures (RM) ANOVA with Tukey's post hoc test, $n = 6$ animals. **d** Statistical plots showed the delta object recognition index under different conditions in control group (mice that expressed GFP in the mPFC). one-way repeated-measures (RM) ANOVA with Tukey's post hoc test, $n = 5$ animals. **e** Statistical plots showed the traveling distance during the test session under different conditions in experimental group (mice that expressed ChR2 in the mPFC), one-way repeated-measures (RM) ANOVA with Tukey's post hoc test, control vs SI -activation, $p = 0.999995$; control vs SUM-activation, $p = 0.966536$; control vs. VTA-activation, $p = 0.000002$; SI-activation vs. SUM-activation, $p = 0.96159$; SI-activation vs VTA-activation, $p = 0.000002$; SUM-activation vs. VTA-activation, $p = 0.000005$. $n = 6$ animals. **f** Statistical plots showed the traveling distance during the test session under different conditions in control group (mice that expressed GFP in the mPFC), one-way repeated-measures (RM) ANOVA with Tukey's post hoc test, $n = 5$ animals. Scale bars in a are 500 μm. MOs secondary motor area, ACA anterior cingulate area, PL prelimbic area, SI substantia innominata, MA magnocellular nucleus, HDB horizontal diagonal band, OT olfactory tubercle, SNR substantia nigra, reticular part; SUM supramammillary nucleus, MM medial mammillary nucleus, VTA ventral tegmental area, IPN interpeduncular nucleus, TM tamoxifen. All data are listed as the Mean ± SEM.

the input patterns to the ET neurons in the mPFC in 5×FAD mice may also show an age-dependent alteration. To test this hypothesis, we performed rabies tracing in Fezf2-CreER mice and 5×FAD$_{Fezf2-CreER}$ mice in both two months of age and six months of age. First, we injected the Cre dependent AAV helper virus into the mPFC of Fezf2-CreER mice or 5×FAD$_{Fezf2-CreER}$ mice at the age of one and half months or five months and induced Cre recombination with tamoxifen three days after the virus injection (Supplementary Fig. 12 and 13a). 21 days later, we injected the RV-EnvA-DG-GFP into the same area. Ten days later, we found massive GFP labeled neurons at the injection site (Supplementary Fig. 12a, b, 13a–c). The dual-color labeled neurons at the injection site were defined as starter cells[55]. Many brain areas were also labeled by RV (Supplementary Fig. 12c and 13d), such as the anterior olfactory nucleus, secondary motor cortex, anterior cingulate cortex, thalamic nuclei, basolateral amygdala, lateral hypothalamic area, hippocampus, and VTA, which was consistent with the brain areas that target GABAergic neurons in the mPFC[55]. To compare the differences of whole-brain inputs between AD$_{Fezf2-CreER}$ mice and Fezf2-CreER mice, we quantified the input neurons and calculated their proportions in each brain area (Supplementary Fig. 12d and 13e). We found that at both two months of age and six months of age, the proportions in most of the input brain areas were similar between AD$_{Fezf2-CreER}$ mice and Fezf2-CreER mice (Supplementary Fig. 12d and 13e). The total number of input neurons and the input neurons/starter cell ratio were also similar between Fezf2-CreER group and AD$_{Fezf2-CreER}$ group at the same age (Supplementary Fig. 12e–h, 13f).

We also tested the specificity of the AAV helper virus by performing control experiments. We injected the Cre dependent AAV-DIO-TVA-mCherry into the vertical diagonal band of ChAT-Cre mice, the RV-EnvA-DG-GFP was injected into the mPFC. Ten days after virus injection, we found that only neurons in the vertical diagonal band were labeled by RV, and most of the labeled neurons were cholinergic neurons (Supplementary Fig. 14a–d). These results validated that the AAV helper virus that expressed TVA and RV were highly specific. Furthermore, to test the specificity of the AAV that expressed RG, we injected the Cre dependent AAV helper virus and the RV into the mPFC of the C57 mouse brains. Ten days after virus injection, we found that only a few neurons in the injection site were labeled by RV, and no long-range input neurons in other brain areas were labeled by RV (Supplementary Fig. 14e–h). Taken together, these results demonstrated the specificity of the virus for monosynaptic tracing.

**Decreased inputs from cholinergic neurons in the horizontal diagonal band to ET neurons in the mPFC of 5×FAD mice.** Previous studies have shown the cholinergic neurons degenerate during AD development[28,39]. Whether the cholinergic inputs to ET neurons in the mPFC also degenerated in the early stage of AD remained unknown. Therefore, we used immunochemical staining against choline acetyltransferase to identify the cholinergic inputs. Previous studies have shown that the diagonal band provided the major cholinergic inputs to the mPFC[55,57]. We found that cholinergic inputs from the basal forebrain to the ET neurons in the mPFC showed no differences between Fezf2-CreER group and AD$_{Fezf2-CreER}$ group at two months of age (Supplementary Fig. 15). However, in the horizontal diagonal band and magnocellular nucleus (HDB-MA) but not the vertical diagonal band and SI of 5×FAD mice, fewer cholinergic neurons were labeled by retrograde RV tracing in AD$_{Fezf2-CreER}$ mice compared to those in the Fezf2-CreER mice at six months of age (Fig. 5a, b). On the other hand, immunochemical staining showed the number of cholinergic neurons in the basal forebrain was similar between C57 mice and 5×FAD mice (Fig. 5c), indicating that no cholinergic neuron loss happened at the early stage of AD, which was consistent with a previous study[39]. These results indicated that the synaptic connections from cholinergic neurons in the HDB-MA to ET neurons in the mPFC were degenerated at six months of age. To further conform our retrograde tracing results, we performed anterograde tracing by injecting Cre dependent AAV virus expressing GFP into the horizontal diagonal band of AD$_{ChAT-Cre}$ mice (5×FAD mice inbred with ChAT-Cre mice) and ChAT-Cre mice at five months of age (Fig. 5d, e). The mPFC was innervated by massive cholinergic axon fibers (Fig. 5f). In 5×FAD mice but not in C57 mice, there were many dystrophic cholinergic axons in the mPFC as described in previous studies[58,59] (Fig. 5g, h). The dystrophic cholinergic axons were close to Aβ plaques (Fig. 5h). We also performed the immunohistochemical staining against choline acetyltransferase at the mPFC. The staining results showed that at six months of age but not two months of age, there were massive dystrophic cholinergic axons within the mPFC of 5×FAD mice (Fig. 5i, j). These dystrophic cholinergic axons indicated that these axons were degenerating[60], which was similarly observed in a previous study[39]. Taken together, our retrograde tracing and anterograde tracing results indicated that the cholinergic inputs from the HDB-MA to ET neurons in the mPFC were degenerating during AD development.

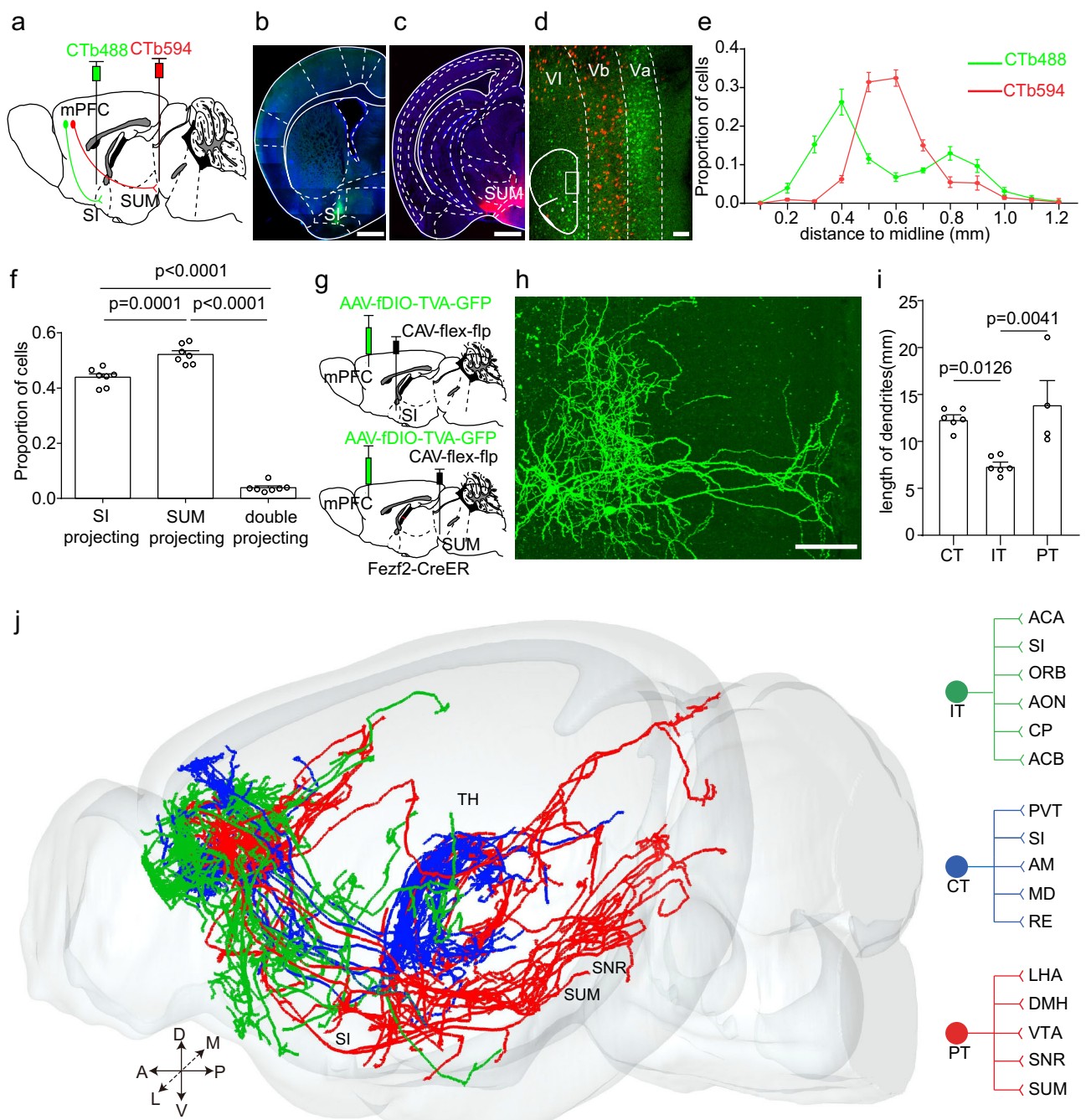

**Fig. 4 SI and SUM projection neurons in the mPFC are anatomically different. a** The strategy to label mPFC-SI projection neurons and mPFC-SUM projection neurons with CTb. **b**, **c** The injection site of CTb in SI and SUM. **d** An example image showed the distribution of CTb labeled neurons in the mPFC. **e** Quantification of the distribution of different CTb labeled neurons in mPFC along the depth of the cortical layers. $n = 7$ animals. **f** Quantification of the percentage of different CTb labeled neurons in the mPFC, $n = 7$ animals. one-way repeated-measures (RM) ANOVA with Tukey's post hoc test. SI projecting vs double projecting, $p = 3.1\times10^{-14}$; SUM projecting vs double projecting, $p = 2.7\times10^{-14}$. **g** The experimental strategy of sparse labeling of the morphology of PFC-SI projection neurons and PFC-SUM projection neurons with AAV. **h** An example image showed the sparse labeled neurons in the mPFC. **i** Quantification of the dendritic length of different types of neurons. one-way repeated-measures (RM) ANOVA with Tukey's post hoc test, CT neurons, $n = 6$; IT neurons, $n = 6$; PT neurons, $n = 4$. **j** The whole-brain axon projection of different types of neurons in the mPFC and their main target brain areas. Scale bars in **b**, **c** are 500 μm. Scale bar in d is 100 μm. Scale bar in **h** is 100 μm. A anterior, D dorsal. L, lateral, M medial, P posterior, V ventral, SI substantia innominata, SUM supramammillary nucleus, IT intratelencephalic, PT pyramidal tract, CT corticothalamic, ACA anterior cingulate area, ORB orbital cortex, AON anterior olfactory nucleus, CP, caudate putamen, ACB nucleus accumbens, PVT paraventricular thalamic nucleus, AM anteromedial thalamic nucleus, MD mediodorsal thalamic nucleus, RE reuniens thalamic nucleus, LHA lateral hypothalamic area, DMH dorsomedial hypothalamic nucleus, VTA ventral tegmental area, SNR substantia nigra, reticular part. All data are listed as the Mean ± SEM.

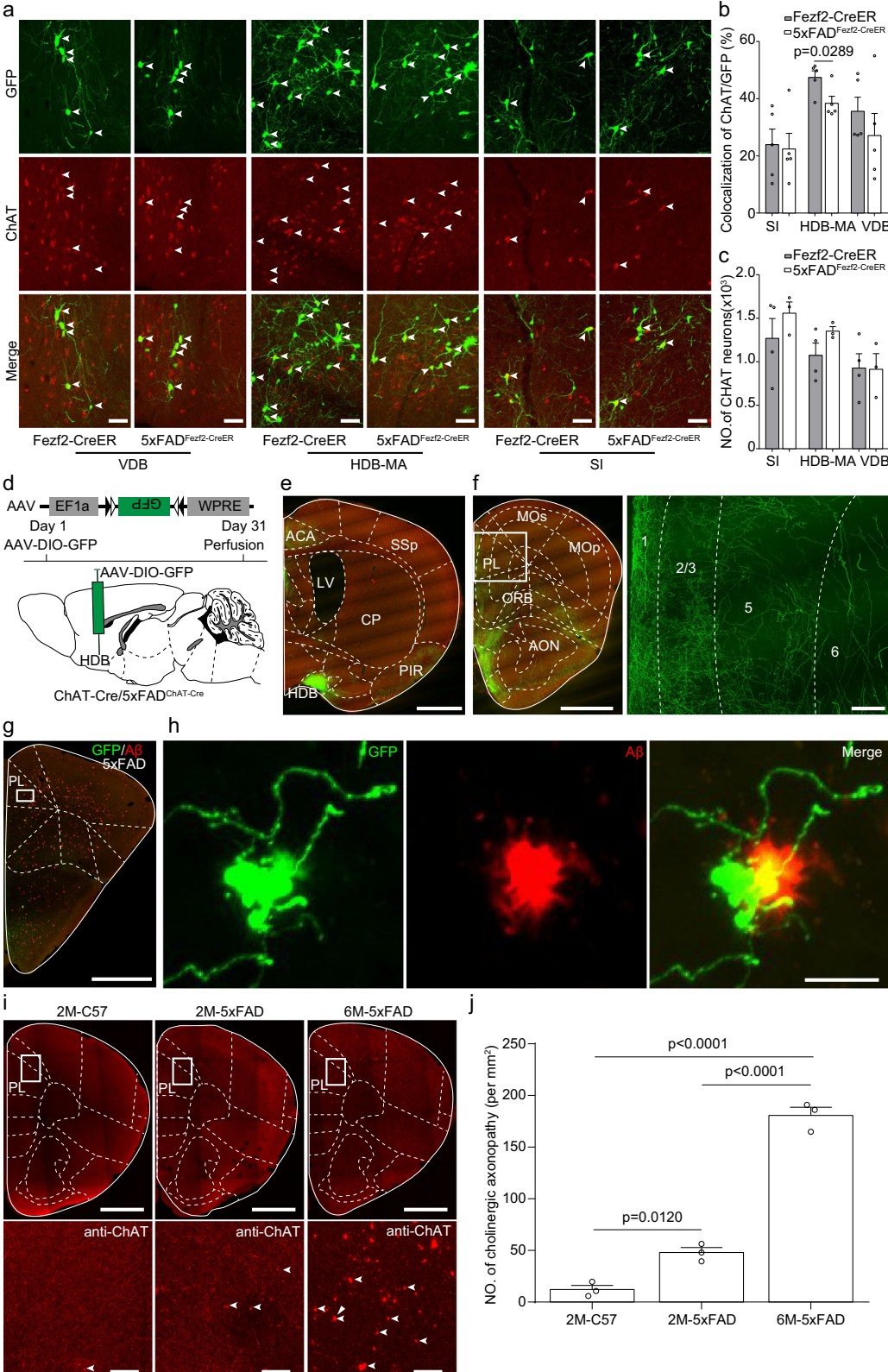

**Functional acetylcholine deficiency to ET neurons in the mPFC of 5×FAD mice**. Our RV tracing results indicated that the ET neurons in the mPFC of 5×FAD mice received fewer cholinergic inputs compared to those of wild type mice. However, the RV tracing mainly reflected the structural alteration, we still need to further investigate whether the functional connectivity between cholinergic neurons and ET neurons in the mPFC was

also altered in 5×FAD mice. A previous study has developed acetylcholine indicators to monitor the in vivo acetylcholine response[61]. We used the indicators to monitor the acetylcholine response in ET neurons in the mPFC. Briefly, a Cre dependent AAV expressing acetylcholine indicators Ach2.0 was injected into the mPFC of Fezf2-CreER mice or AD$_{\text{Fezf2-CreER}}$ mice, and we induced Cre recombination with tamoxifen (Fig. 6a). One month

**Fig. 5 Decreased cholinergic inputs from horizontal diagonal band to extratelencephalic projection neurons in the mPFC of 5×FAD mice. a** Immunostaining against ChAT (red) to identify the rabies labeled cholinergic input neurons (green) from vertical diagonal band, horizontal diagonal band and SI to extratelencephalic projection neurons in the mPFC. **b** Comparison of rabies virus labeled ChAT neurons between AD$_{Fezf2-CreER}$ and Fezf2-CreER mice in vertical diagonal band, horizontal diagonal band, and SI brain area ($n = 5$ animals for AD, $n = 5$ animals for WT). All error bars represent Mean ± SEM and the significant differences were indicated by $p$ value (two-tailed unpaired $t$ test). **c** Comparison of the number of cholinergic neurons between AD$_{Fezf2-CreER}$ and Fezf2-CreER mice in the vertical diagonal band, horizontal diagonal band, and SI brain area (two-tailed unpaired $t$ test, $n = 3$ animals for AD, $n = 4$ animals for WT). **d** Labeling the output of cholinergic neurons in the horizontal diagonal band of AD$_{ChAT-Cre}$ mice or ChAT-Cre mice. **e** The virus injection site at the horizontal diagonal band. **f** The cholinergic axon terminals in the mPFC. **g** The cholinergic axon terminals and the Aβ plaques in the mPFC of AD$_{ChAT-Cre}$ mice. **h** Enlarged image boxed in **g** showed the degenerated cholinergic axons in the mPFC. **i** Representative images showed the cholinergic axonopathy in the mPFC of 5×FAD mice at different ages. **j** Quantification of the cholinergic axonopathy in the mPFC of 5×FAD mice at different ages. $n = 3$ animals for each group. one-way repeated-measures (RM) ANOVA with Tukey's post hoc test. 2M-C57 vs. 6M-5×FAD, $p = 0.000002209$; 2M-5×FAD vs 6M-5×FAD, $p = 0.000009272$. Scale bars in **e, f** (left) and **g** are 1 mm. Scale bar in **f** (right) is 100 μm. Scale bars in a and **h** are 50 μm. Scale bars in **g, l** (upper) are 1 mm. Scale bars in **l** (lower) are 50 μm. SI substantia innominata, HDB horizontal diagonal band, MA magnocellular nucleus, ACA anterior cingulate area, CP caudate putamen, PIR piriform cortex, LV lateral ventricle, SSp primary somatosensory area, PL prelimbic area, ORB orbital cortex, AON anterior olfactory nucleus, Mop primary motor area, MOs secondary motor area. All data are listed as the Mean ± SEM.

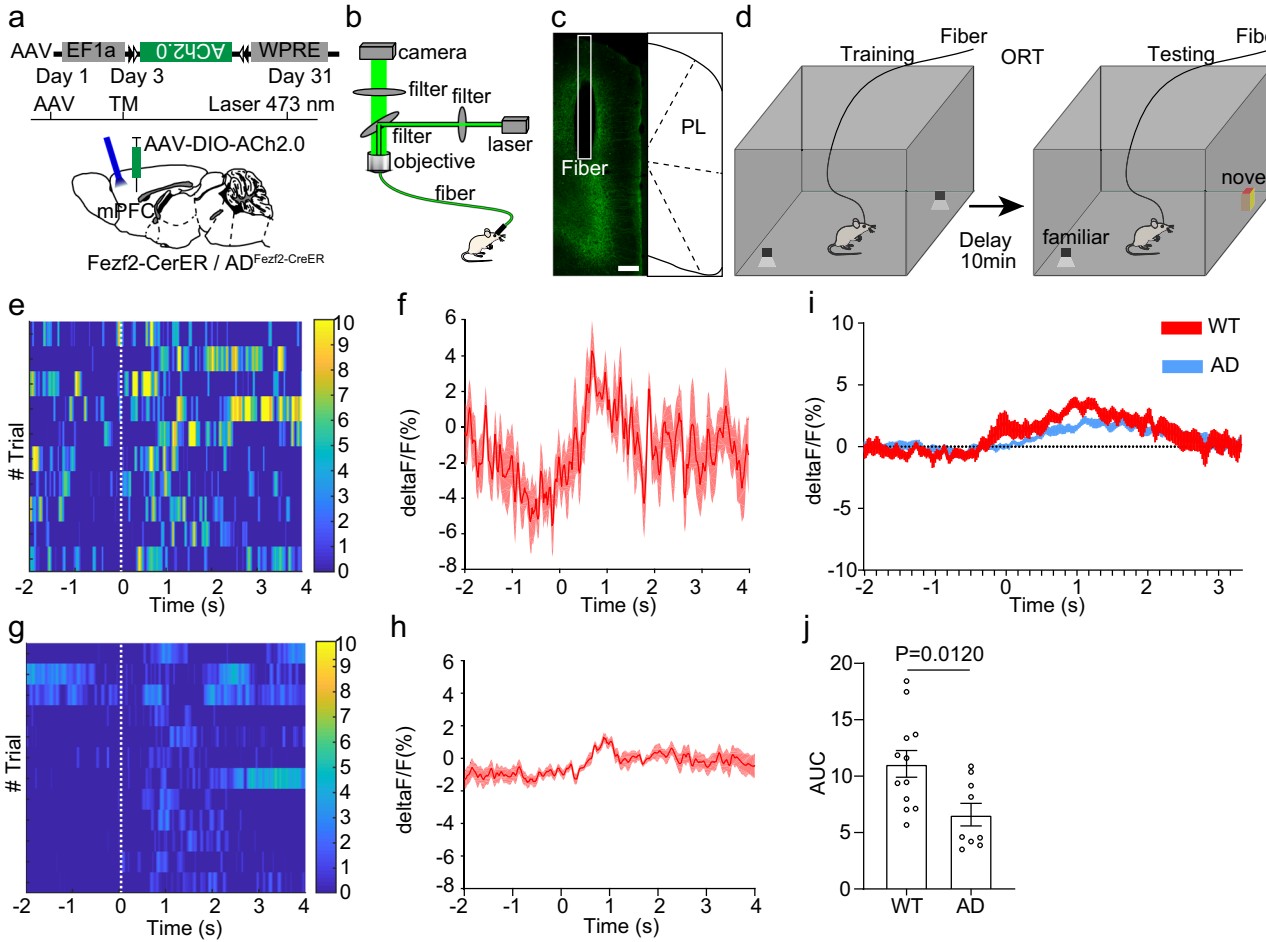

**Fig. 6 Functional acetylcholine deficiency to extratelencephalic projection neurons in the mPFC of 5×FAD mice. a** Expressing the acetylcholine indicators in Fezf2 positive neurons in the mPFC with a Cre dependent virus. The Cre expression was induced by tamoxifen 3 days after the virus injection. **b** The scheme of fiber photometry. **c** The expression of acetylcholine indicators in Fezf2 positive neurons in the mPFC. **d** The experimental design of object recognition test. **e** Heatmap of acetylcholine response of extratelencephalic projection neurons in the PL of Fezf2-CreER mice during object interactions. **f** Average plots across different trials of individual animals of acetylcholine response of extratelencephalic projection neurons in PL of Fezf2-CreER mice during object interactions. **g** Heatmap of acetylcholine response of extratelencephalic projection neurons in PL of AD$_{Fezf2-CreER}$ mice during object interactions. **h** Average plots across different trials of individual animals of acetylcholine response of extratelencephalic projection neurons in PL of AD$_{Fezf2-CreER}$ mice during object interactions. **i** Average plots of acetylcholine response of extratelencephalic projection neurons in the PL of both Fezf2-CreER mice and AD$_{Fezf2-CreER}$ mice during multiple tests on multiple animals (Fezf2-CreER, $n = 12$ animals; AD$_{Fezf2-CreER}$ mice, $n = 9$ animals). **j** The quantification of AUC in **i** The acetylcholine response of extratelencephalic projection neurons in PL of both Fezf2-CreER mice is stronger than that in AD$_{Fezf2-CreER}$ mice. (Two-tailed Mann-Whitney test, Fezf2-CreER mice, $n = 12$ animals; AD$_{Fezf2-CreER}$ mice, $n = 9$ animals.) Scale bar in **c** is 200 μm. ACA anterior cingulate area, PL prelimbic area, TM tamoxifen. All data are listed as the Mean ± SEM. The color bars in the figure indicated df/f (%).

later, we assigned the mice to NOR and performed fiber photometry to monitor the acetylcholine response in ET neurons in the mPFC (Fig. 6b–d). During object interaction, the ET neurons in the mPFC of Fezf2-CreER mice showed obvious acetylcholine response (Fig. 6e, f). However, the acetylcholine response in ET neurons in the mPFC of AD$_{Fezf2-CreER}$ mice was decreased compared to that of Fezf2-CreER mice (Fig. 6g, h). The overall acetylcholine response in ET neurons in the mPFC of Fezf2-CreER mice was stronger than that in the AD$_{Fezf2-CreER}$ mice (Fig. 6i, j). These results revealed that the functional acetylcholine deficiency to ET neurons in the mPFC of 5×FAD mice.

**Manipulation of the activity of cholinergic axon fibers in the mPFC affected object recognition memory**. Our results have shown that the ET neurons in the mPFC of 5×FAD mice received fewer cholinergic inputs from the horizontal diagonal band. So, we would like to investigate whether increasing acetylcholine release in the mPFC of 5×FAD mice could rescue the ORM. To test this hypothesis, we expressed ChR2 in the cholinergic neurons in the horizontal diagonal band of AD$_{ChAT-Cre}$ mice at five months of age with AAV (Fig. 7a). Immunochemical staining against choline acetyltransferase validated the specificity of AAV virus (Fig. 7b). Most of the neurons expressing ChR2 were restricted to cholinergic neurons (795/805, n = 6 mice). We assigned the test mice to NOR and activated the cholinergic fibers in the mPFC with 473 nm laser (10 Hz, 15 ms pulse width, 3.5-5 mW at the tip of the optical fiber). Compared to the tests without light stimulation, light stimulation of cholinergic fibers that expressed ChR2 in the mPFC significantly increased exploration time with novel object of 5×FAD mice, and they showed novel object preference (Fig. 7c–e), which indicated that activation of cholinergic fibers in the mPFC of 5×FAD mice did improve their ORM. Besides, activation of cholinergic fibers in the mPFC did not affect locomotion (Fig. 7e). We also performed control experiments in which test mice expressed GFP instead of ChR2 in the cholinergic neurons in the horizontal diagonal band. Light stimulation of GFP positive cholinergic axon fibers in the mPFC did not affect the object recognition or locomotion (Fig. 7f–h).

Next, we tested the necessity and sufficiency of acetylcholine release in the mPFC of ChAT-Cre mice for ORM. We expressed NpHR in the cholinergic neurons in the horizontal diagonal band of ChAT-Cre mice with AAV (Supplementary Fig. 16a, b). The immunochemical staining against ChAT revealed the specificity of the AAV that expressed NpHR (1270/1359, $n = 4$ mice, Supplementary Fig. 16b). We assigned the test mice to NOR and inhibited the cholinergic fibers in the mPFC with 570 nm laser (3.5–5 mW at the tip of the optical fiber, insistent inhibition) at different phases of the NOR. Inhibition of cholinergic axon fibers in the mPFC of ChAT-Cre mice during training and delay phase did not change the preference of these mice towards novel object (Supplementary Fig. 16c, d). These results indicated that inhibition of acetylcholine release during training and delay phase did not affect the ORM expression. However, compared to the tests without light inhibition, light inhibition of cholinergic axon fibers that expressed NpHR in the mPFC during the test phase significantly decreased exploration time with the novel object of ChAT-Cre mice (Supplementary Fig. 16e), which indicated that the ORM of those mice was impaired. Inhibition of cholinergic fibers in mPFC also did not affect the locomotion ability of the mice (Supplementary Fig. 16f). These results implied that cholinergic inputs to mPFC are necessary for normal ORM expression. Previous studies also showed that overstimulation of M1 receptor in the mPFC also could cause short-term memory impairment[35,36]. To examine whether activation of cholinergic

fibers in the mPFC of ChAT-Cre mice impaird ORM, we expressed ChR2 in the cholinergic neurons in the horizontal diagonal band of ChAT-Cre mice (Supplementary Fig. 17a). We assigned the test mice to NOR and activated the cholinergic fibers in the mPFC with 473 nm laser (Supplementary Fig. 17b). Activation of cholinergic axon fibers in the mPFC of ChAT-Cre mice during training and delay phase did not change the preference of the mice towards novel objects (Supplementary Fig. 17c, d). Compared to the tests without light activation, light stimulation of cholinergic fibers that expressed ChR2 in the mPFC during test phase significantly decreased the exploration time with novel objects of ChAT-Cre mice (Supplementary Fig. 17e), which indicated that activation of cholinergic fibers in mPFC of ChAT-Cre mice did impair ORM expression, these results were consistent with previous studies[35,36]. Activation of cholinergic fibers in the mPFC also did not affect the locomotion of the animals (Supplementary Fig. 17f).

**Acetylcholine deficiency altered the response pattern of ET neurons in the mPFC to familiar objects**. Since activation of cholinergic axon fibers in the mPFC of 5×FAD mice could effectively improve the ORM, we would like to explore how acetylcholine release in the mPFC affected the activities of the ET neurons. To achieve this goal, we designed an experimental protocol to combine fiber photometry and chemogenetics together. To activate the cholinergic neurons in the horizontal diagonal band that project to the mPFC of 5×FAD mice, we injected Cre dependent CAV that expressed Flp into the mPFC of AD$_{ChAT-Cre}$ mice at five months old. A Flp dependent AAV that expressed excitatory chemogenetics receptor hM3Dq and mCherry was injected into the horizontal diagonal band. To simultaneously record the activities of ET neurons in the mPFC, a retroAAV that expressed GCaMPs was injected into the SUM to label the ET neurons in the mPFC (Supplementary Fig. 18a–c). To test the specificity of the AAV virus that expressed hM3Dq, the CAV and AAV was also injected into the ChAT-Cre: Ai47 mouse brain. Three weeks after virus injection, we found that most of the neurons that expressed mCherry also expressed GFP (Supplementary Fig. 18d). Thus, our results demonstrated the specificity of the AAV. To mimic the acetylcholine deficiency in the 5×FAD mouse brain, we injected a Cre dependent AAV that expressed inhibitory chemogenetics receptor hM4Di and mCherry into the horizontal diagonal band of ChAT-Cre mice at five months of age (Supplementary Fig. 18a–c). A retroAAV that expressed GCaMPs was also injected into the SUM to label the ET neurons in the mPFC for fiber photometry. The specificity of the AAV that expressed hM4Di and mCherry was also tested with the injection into the mouse brain of ChAT-Cre: Ai47 (Supplementary Fig. 18e). We also examined the location of the injection site at the horizontal diagonal band. We found that most of the virus infected neurons were confined to the horizontal diagonal band (Supplementary Fig. 18f–i), which assured that our chemogenetic manipulation is brain region specific. One week after virus injection, drinking water containing CNO was provided to the mice to activate the chemogenetic receptors. The administration of CNO lasted for a week, and then, the fiber photometry was performed. To examine whether the chemogenetic receptors worked properly in vivo, we sacrificed the tested animals after fiber photometry and performed c-fos staining. We found that in the hM3Dq+CNO group, most of the mCherry+ neurons colocalized with the c-fos positive neurons (Supplementary Fig 18j, k), while only a few mCherry positive neurons colocalized with the c-fos positive neurons in hM4Di+CNO group (Supplementary Fig. 18j, k). To explore whether the administration of CNO alone has any effect on the behavior performance, we also performed

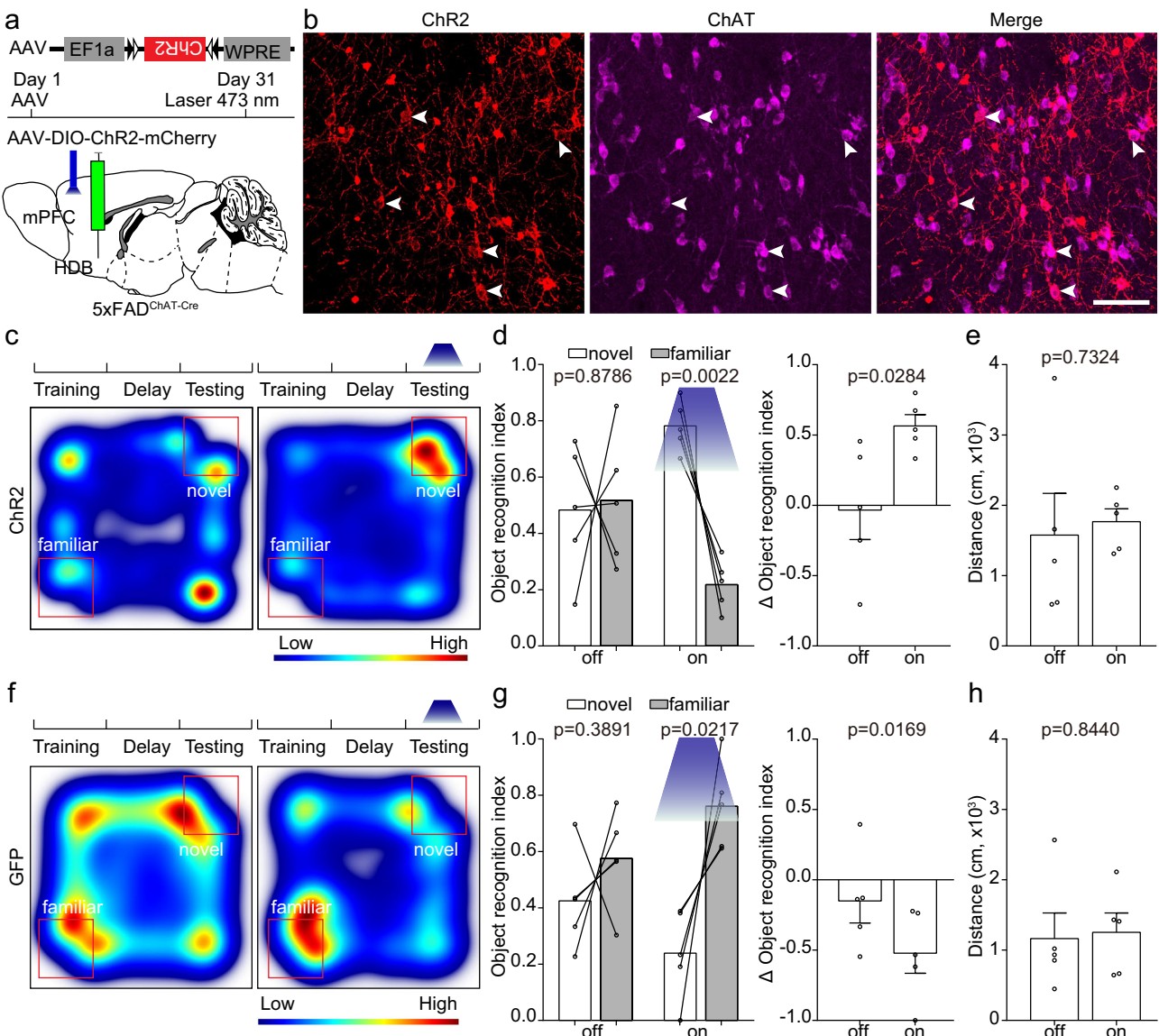

**Fig. 7 Activation of cholinergic axon terminals in the mPFC of 5×FAD mice improved object recognition memory expression. a** The experimental strategy of activation of cholinergic terminals in the mPFC. **b** Immunostaining against ChAT to validate the specificity of the AAV. **c** Heat-map plots showed object recognition memory measures from the object recognition test. Red = more time, blue = less time. The heatmap in the left and right showed examples of AD$_{ChAT-Cre}$ mice that expressed ChR2 in the cholinergic terminals in the mPFC exploring the familiar object and novel object during test session without or with light stimulation, respectively. **d** Statistical plots of object recognition index and delta object recognition index in laser off and laser on sessions. (left and right, two-tailed paired *t* test). $n = 5$ animals. **e** The statistical plot showed the traveling distance of AD$_{ChAT-Cre}$ mice with or without light activation of the cholinergic terminals in the mPFC. two-tailed paired *t* test, $n = 5$ animals. **f** The heatmap in the left and right showed examples of AD$_{ChAT-Cre}$ mice that expressed GFP in the cholinergic terminals in the mPFC exploring the familiar object and novel object during test session without or with light stimulation, respectively. **g** Statistical plots of object recognition index and delta object recognition index in laser off and laser on sessions. (left and right, two-tailed paired *t* test). $n = 5$ animals. **h** Statistical plots showed the traveling distance in GFP group with or without light stimulation. two-tailed paired *t* test, $n = 5$ animals. Scale bar in b is 50 μm. All data are listed as the Mean ± SEM.

the control experiment on the CNO administration. The 5×FAD mice were divided into two groups. One group of animals were given drinking water with CNO while the other group of animals were given ordinary water. After seven days of drug delivery, the animals were tested in NOR. The results showed that the animals in both groups exhibit similar exploration time towards novel and familiar objects, indicating that CNO administration alone cannot alter the behavior performance of the animals in NOR (Supplementary Fig. 18l).

In the control (C57) group, retroAAV that expressed GCaMPs was also injected into the SUM to label the ET neurons in the

mPFC for fiber photometry. Three weeks after the virus injection, we performed fiber photometry to record the response of ET neurons in the mPFC to both novel and familiar objects (Fig. 8a, b, Supplementary Fig. 18m). In C57 group, the response of ET neurons in the mPFC to familiar object was stronger than that to novel objects (Fig. 8c). These results indicated that the ET neurons in the mPFC played a role of novel/familiar object discriminator. In 5×FAD group, the response of ET neurons in the mPFC to novel objects remained unaltered (Supplementary Fig. 19a). However, the response of ET neurons in the mPFC to familiar objects decreased significantly (Fig. 8d, Supplementary Fig. 19a). The response of ET

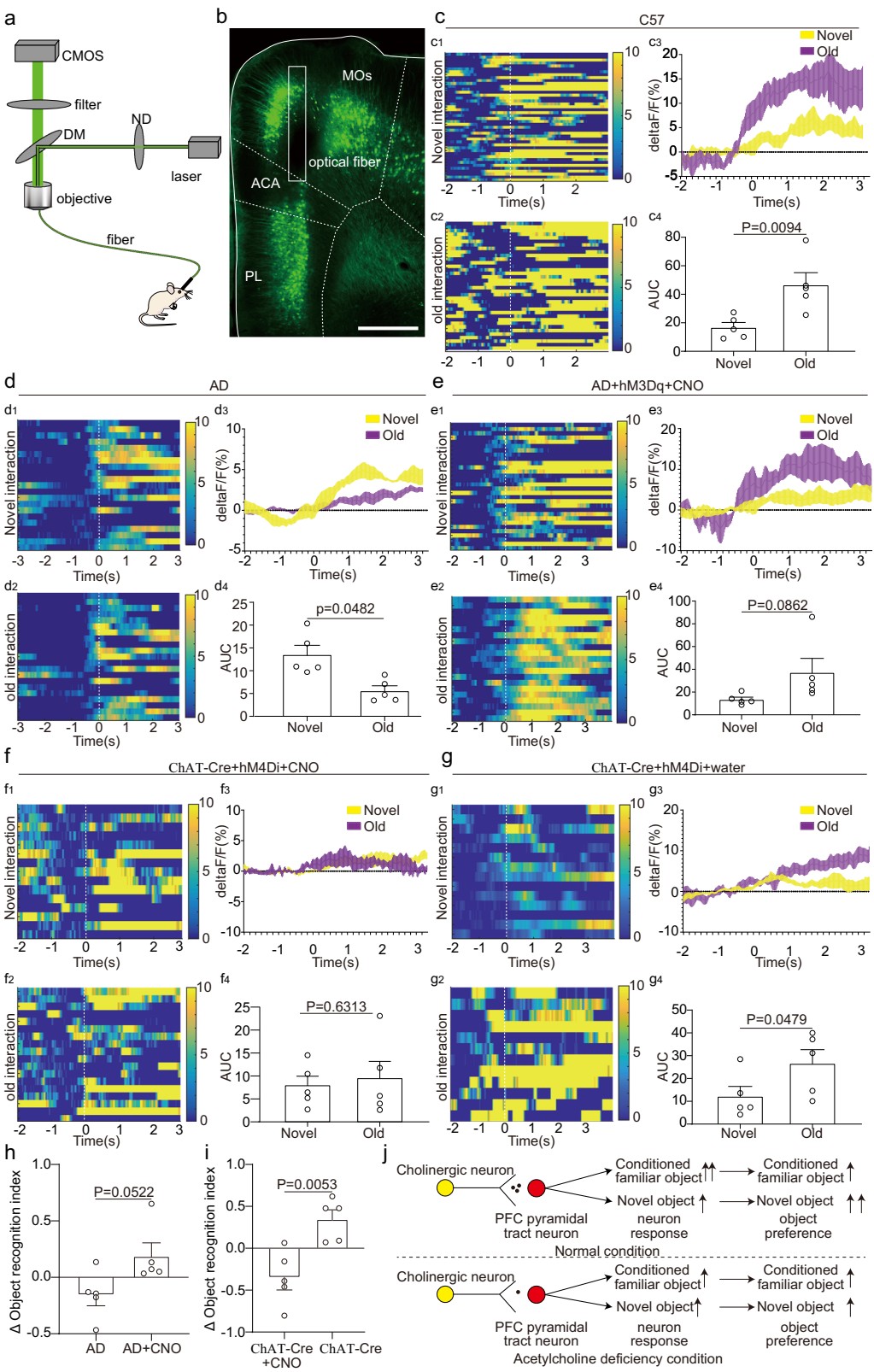

neurons in the mPFC to familiar objects was slightly correlated with the object recognition ability of the animals (Supplementary Fig. 19b). In hM3Dq+CNO group, the response patterns of ET neurons in the mPFC of 5×FAD mice to novel and familiar objects was restored via activation of cholinergic neurons in the horizontal diagonal band (Fig. 8e). In hM4Di+CNO group, the inhibition of cholinergic neurons in the horizontal diagonal band abolished the

stronger response of the ET neurons in the mPFC of ChAT-Cre mice to familiar object (Fig. 8f). To further test the necessity of the CNO for driving the chemogenetic receptors, we replaced the water containing CNO with ordinary drinking water in hM4Di+CNO group. One day after the replacement, the fiber photometry was performed again. The withdrawal of CNO restored the normal response patterns of ET neurons in the mPFC. The ET neurons

**Fig. 8 Acetylcholine deficiency to extratelencephalic projection neurons in the mPFC altered the response patterns of extratelencephalic projection neurons in the mPFC to novel object and familiar object. a** The experimental strategy of fiber photometry. **b** The position of optical fibers in the mPFC for fiber photometry. **c** The response of extratelencephalic projection neurons in the mPFC of C57 mice to familiar object and novel object. c1-c3, Heatmap of calcium response of individual animal and average response plots of extratelencephalic projection neurons in the mPFC of different C57 mice. c4, Quantification of AUC in c3. $n = 5$ animals. **d** The response of extratelencephalic projection neurons in the mPFC of 5×FAD mice to familiar object and novel object. d1-d3, Heatmap of calcium response of individual animal and average response plots of extratelencephalic projection neurons in the mPFC of different 5×FAD mice. d4, Quantification of AUC in d3. $n = 5$ animals. **e** The response of extratelencephalic projection neurons in the mPFC of 5×FAD mice to familiar object and novel object during the activation of cholinergic neurons in the horizontal diagonal band with chemogenetics. e1-e3, Heatmap of calcium response of individual animal and average response plots of extratelencephalic projection neurons in the mPFC of different 5×FAD mice with cholinergic activation. e4, Quantification of AUC in e3. $n = 5$ animals. **f** The response of extratelencephalic projection neurons in the mPFC of ChAT-Cre mice with cholinergic neuron inhibition via chemogenetics. f1-f3, Heatmap of calcium response of individual animal and average response plots of extratelencephalic projection neurons in the mPFC of different ChAT-Cre mice with cholinergic inhibition. f4, Quantification of AUC in f3. $n = 5$ animals. **g** The response of extratelencephalic projection neurons in the mPFC of ChAT-Cre mice (control group). g1-g3, Heatmap of calcium response of individual animal and average response plots of extratelencephalic projection neurons in the mPFC of different ChAT-Cre mice. g4, Quantification of AUC in g3. $n = 5$ animals. **h** Delta object recognition index comparison between 5×FAD$_{ChAT-Cre}$ group and 5×FAD$_{ChAT-Cre}$ chemogenetics activation group. $n = 5$ animals. **i** Delta object recognition index comparison between ChAT-Cre group and ChAT-Cre chemogenetics inhibition group. $n = 5$ animals. **j** The scheme of how acetylcholine regulates the response of extratelencephalic projection neurons in the mPFC to novel object and familiar object. **c, d, f, g,** two-tailed paired t test. **e** two-tailed Wilcoxon matched-pairs signed-rank test. **h** two-tailed Mann-Whitney test; **i,** two-tailed unpaired *t* test. Scale bar in **b** is 500 μm. MOs, secondary motor area; ACA, anterior cingulate area; PL, prelimbic area. All data are listed as the Mean ± SEM. The color bars in the figure indicated df/ f (%).

---

showed stronger response to familiar objects than to novel objects (Fig. 8g). The chemogenetic manipulation not only modulated the activity of the ET neurons in the mPFC, but also altered the ORM of the test mice. Compared to the 5×FAD group, the mice in hM3Dq+CNO group showed obvious preference towards the novel object (Fig. 8h), indicating improvement of ORM. On the other hand, compared to the hM4Di+CNO group, the withdrawal of CNO also restored the preference towards the novel object (Fig. 8i). These results were consistent with our optogenetic manipulation of cholinergic axon fibers in the mPFC (Fig. 7 and Supplementary Fig. 16). Thus, according to our fiber photometry and behavior test results, we came up with a circuit mechanism of how acetylcholine release in the mPFC modulated the response patterns of ET neurons to different objects (Fig. 8j): Under normal condition, the ET neurons in the mPFC showed stronger response to familiar objects, and the mice showed preference to novel object. Under the acetylcholine deficiency condition, the stronger response to the familiar object of the ET neurons in the mPFC was abolished, and the ORM of the mice was impaired.

To directly test that the biased response of ET neurons in the mPFC towards familiar and novel objects are essential for object recognition, we also performed the light inhibition of ET neurons in the mPFC of Fezf2-CreER mice only when the animals were exploring the familiar and novel objects. Unlike our previous light manipulation in which the light was delivered in the whole arena, we only delivered the light to inhibit the ET neurons in the mPFC when animals entered the zone where the familiar and novel objects were located (Supplementary Fig. 20a, b). We found that the light inhibition at the object zone also impaired the object recognition of the animals (Supplementary Fig. 20c–e). These results indicated that the biased response of ET neurons in the mPFC towards familiar and novel objects are essential for ORM.

The biased response of ET neurons in the mPFC and the different preferences towards novel and familiar objects brought up the possibility that the ET neurons in the mPFC may encode place preference to express the preference towards novel objects rather than directly encode ORM. To test whether ET neurons in the mPFC can encode place preference, we performed intraCranial light administration in a specific subarea (iClass) test. In the iClass training sessions, mouse exploration within a marked center area of an open field triggered the delivery of light stimulation to the ET neurons in the mPFC which expressed ChR2 or NpHR through an implanted optical fiber. Normally,

mice would avoid the center area and prefer areas near the walls and corners. If the ET neurons in the mPFC can directly encode place preference or aversion, the manipulation of ET neurons in the mPFC may change the activity patterns of the test mice. During two sessions of training, activation or inhibition of ET neurons in the mPFC did not induce robust place preference or aversion (Supplementary Fig. 21). These results indicated that the ET neurons in the mPFC do not directly encode reward signals. The biased response of ET neurons in the mPFC and the different preferences towards novel and familiar objects reflected different memory retrieval rather than place preference.

## Discussion

In this study, we compared the activities and whole-brain inputs of ET neurons in the mPFC between 5×FAD mice and wild type mice. The ET neurons in the mPFC of 5×FAD mice showed decreased responses during the NOR. Activation of these neurons or their axons in the SUM could rescue ORM impairment in 5×FAD mice while inhibition of them or their axons in SUM could impair ORM expression but not formation and consolidation in wild type mice. Whole-brain RV screening and in vivo recordings with acetylcholine indicators revealed that ET neurons in the mPFC of 5×FAD mice received fewer cholinergic inputs from the horizontal diagonal band. Activation of cholinergic fibers in the mPFC could rescue ORM impairment in 5×FAD mice while inhibition of cholinergic fibers in the mPFC could impair ORM expression but not formation and consolidation in wild type mice. The ET neurons in the mPFC played a role as the object discriminator in normal conditions, and acetylcholine deficiency abolished the ability of object discrimination of ET neurons in the mPFC.

PFC dysfunction and ORM impairment have been observed in AD patients and mouse models of AD[4,37]. However, the circuit mechanism of these impairment remains unclear. Our results revealed that ET neuron dysfunction in the mPFC was responsible for ORM impairment. Many cortical brain regions such as mPFC, hippocampus, entorhinal, perirhinal, and parahippocampal cortex have been shown to be involved in ORM[11], but people have not paid enough attention to which subcortical brain regions are responsible for ORM. A recent study demonstrated that the different neurons in the SUM encoded social and contextual novelty[62]. Our c-fos staining results and optogenetic manipulation results revealed that the SUM is also involved in ORM (Fig. 3 and

Supplementary Fig. 5 and 10). The SUM is a brain area with diverse cell types and complex neuronal connectivities[62–64]. The direct projection from the mPFC to SUM has been identified in a previous study[65]. Our results showed that the direct projection from mPFC to SUM could regulate the ORM. These results revealed how ORM was processed in the brain, which could shed light on the understanding of neural circuitries responsible for ORM and could help researchers understand how the brain process, restore, and express ORM. Further investigation is still needed to explore how the information delivered from the mPFC was processed within the SUM and transmitted to the downstream brain regions. In our results, we found that the ET neurons in the mPFC displayed stronger responses to familiar objects (Fig. 8). This similar response pattern was also observed in the VGAT positive neurons in the SUM[64]. Thus, the connectivity between the ET neurons in the mPFC and neurons in the SUM could potentially be the feedback circuit for ORM transmitting back to hippocampus.

The hippocampus is another brain region responsible for memory formation, consolidation and expression. The role of hippocampus on ORM remains highly debated due to conflicting findings across temporary and permanent hippocampal lesion studies[66]. The inconsistent results may be a consequence of the different time windows of manipulation, different brain regions of manipulation (dorsal hippocampus vs ventral hippocampus), and different delay time of the task. For example, studies have indicated that the ventral hippocampus is essential for spatial memory[64,67] but not for object recognition test[50,64]. The inactivation of dorsal hippocampus impaired the object memory retrieval only in the tasks that contained longer delay time (24 h) rather than a short delay time (5 min)[68,69]. Nevertheless, other studies indicated that the dorsal hippocampus is also essential for ORM with short delay time. Inactivation of dorsal hippocampus impaired ORM formation, consolidation and expression[70,71]. Our study together with previous studies found that mPFC is essential for ORM expression[27] (Supplementary Fig. 7). In our results, we found that inhibition of the ET neurons in the mPFC only impaired ORM expression but not formation and consolidation (Supplementary Fig. 7 and 8). Therefore, it is likely that the ORM is encoded and consolidated in dorsal hippocampus and other brain areas. Then, the memory information transmitted to downstream brain areas include mPFC for retrieval[72]. There have been reports showing that altered hippocampal circuits in AD mouse models can lead to abnormal spatial memory and anxiety expression[38,40,73,74], but whether and how alteration of hippocampal circuits caused by AD affect ORM needs to be investigated in future studies.

It is also noteworthy that the selective excitotoxic lesion of mPFC indicated that the mPFC is not essential for ORM[75–77]. However, other studies using drug inhibition of mPFC showed that infusion of glutamate antagonist into the mPFC before the test phase of the NOR could significantly impair the object recognition performance during the test phase[27,78]. This reflected that the time window of manipulation on the mPFC is critical. In lesion studies, the structure lesion occurred before the object recognition test and the test was done weeks or even months after the lesion surgery. The time window between the surgery and the behavior test may allow the brain to adapt to the new neural network to process the novel object information during the behavior test. However, our optogenetic inhibition can control the neuron activity at the millisecond scale, which allows us to more precisely manipulate the activity of specific neural circuits in different phases of the task to understand their potential functions. Furthermore, in monkeys, the prefrontal neurons have been reported to show increases in responses to familiar stimuli compared to novel stimuli[79], which is consistent with our findings in rodent PFC neurons. These results suggested an involvement of the mPFC in single-item recognition memory.

In our results, we found that the decreased response of the ET neurons in the mPFC was correlated with impaired ORM in 5×FAD mice. However, activation of the ET neurons in the mPFC of wild type mice did not improve ORM. These results were consistent with some previous studies[25,26], suggesting that both elevating and lowering the excitation level of the excitatory neurons in the cortex would lead to an excitation/inhibition imbalance and result in cognition defects. In our results, we found that manipulation of ET neurons or cholinergic axon fibers in the mPFC during the test phase instead of the training phase-modulated the objection memory (Supplementary Fig. 7, 8, 16 and 17). These results were consistent with previous studies, which suggested the memory defects in AD were mainly caused by the impairment of memory expression[80]. Interestingly, the MRI study on working memory of human brains indicates that the superficial layers of the PFC mainly responded in the delayed phase of working memory while the deep layers of the PFC mainly responded in the expression phase[81]. These results were consistent with our optogenetical manipulation results. We still need to investigate how different types of pyramidal neurons in the mPFC encoded different phases of working memory in future studies.

In our optogenetic manipulation experiments, we found that activation of ET neurons in the mPFC increased the locomotion of the animals (Fig. 2 and Supplementary Fig. 8). However, our photometry results showed that the locomotion of the animals was not correlated with the neuronal activities of the ET neurons in the mPFC (Supplementary Fig. 4). Moreover, we found that inhibition of ET neurons in the mPFC had no effect on the locomotion of the animals (Supplementary Fig. 7). Therefore, we assumed that the increase of locomotion was most likely to be caused by the unnatural optogenetic activation of the somas in the PFC. Moreover, we found that activation of ET neurons in the AD (AD$_{Fezf2-CreER}$) group and wild-type (Fezf2-CreER) group had similar effect on locomotion but opposite effects on object recognition (Fig.2 and Supplementary Fig. 8). The neuronal activity of ET neurons in the mPFC showed no correlation with the locomotion of the animals (Supplementary Fig. 4). Therefore, the ORM alteration of the animals is unlikely to be related to the locomotion change. In our axon terminal manipulation experiments, we did find that inhibition of the mPFC-VTA pathway decreased the locomotion of the animals. The mPFC-VTA pathway has been implicated to be involved in the social reward process and can be impaired by chronic stress[82]. Therefore, the locomotion effect of inhibition of the mPFC-VTA pathway is more likely due to the emotional state change rather than movement disability. The exact functions of the mPFC-VTA pathway needs to be further investigated.

Cholinergic modulation of cortical functions has been extensively studied[83]. The cholinergic inputs to the mPFC have been indicated to be involved in attentional tasks[84–86]. The excitation of cholinergic receptors in the mPFC can enhance the sensory information input to optimize attention performance[84]. Decreased acetylcholine release in the mPFC severely disrupted attention performance[87]. Acetylcholine deficiency and cholinergic neuron degeneration have been observed in AD brains for decades[28]. Cholinergic fiber lesions have been found in many limbic brain areas, including the hippocampus and the mPFC[39]. Specifically, the cholinergic input defects to layer VI of the mPFC have been identified in an AD mouse model (TgCRND8 model)[88]. However, it is not well known whether other specific types of mPFC neuron received fewer cholinergic inputs during AD development. By applying whole-brain RV tracing and in vivo acetylcholine recordings, we demonstrated that ET

neurons in the mPFC also received fewer cholinergic inputs from the HDB-MA. The decreased cholinergic inputs may cause attention defects in 5×FAD mice and impair the behavior performance. The impaired dendritic spine plasticity in ET neurons in the PFC in mouse models of AD has been observed via chronic two-photon imaging. Impaired circuits plasticity in excitatory neurons in the mPFC and learning ability of 5×FAD mice were also observed via chronic two-photon calcium imaging[16]. While in recent studies, acetylcholine release in cortical areas has been shown to be critical for cortical plasticity and normal learning ability[89]. Therefore, the acetylcholine deficiency to ET neurons in the mPFC could be the main cause of impaired dendritic spine plasticity and thus disrupted learning ability. Previous studies also showed abnormal activities of layer II/III pyramidal neurons and interneurons in the mPFC of AD mouse model (5×FAD mouse model and APPswe/ PS1dE9 mouse model)[90,91]. It is worth exploring whether these neuron types also receive fewer cholinergic inputs and whether the main cause of their abnormal activity is due to the fewer cholinergic inputs to these neuron types.

Previous studies have raised heated discussion about the cholinergic hypothesis of AD[28,92]. Non-pharmacological approaches such as exercise did improve ORM and increased acetylcholine release in the mPFC of the rat model of AD (Aβ injection-induced AD model)[93], which indicated that increasing acetylcholine release in the mPFC did play a role in regulating the ORM of AD animal models or even AD patients. Our results showed that normal acetylcholine release in the mPFC was necessary for ORM (Supplementary Fig. 16). Activation of cholinergic fibers in the mPFC of 5×FAD mice could rescue the ORM impairment (Fig. 7). However, overactivation of cholinergic fibers in the mPFC of wild type mice could also impair ORM (Supplementary Fig. 17), which was similar to the conclusions in some previous studies[35,36]. These results implied that there is a balance of acetylcholine release in the neocortex, and both acetylcholine deficiency and overstimulation could cause short-term memory impairment. Therefore, the dose of procholinergic drugs and muscarinic agonists used for AD treatments should be suitable for maintaining the balance of acetylcholine release in the brain to avoid both acetylcholine deficiency and overstimulation so the procholinergic drugs and muscarinic agonists could maximize the impacts on AD treatment.

Previous studies on AD have focused on the molecular alterations such as Aβ and tau protein accumulation in the brain[94,95]. The neural circuitry alterations in the AD brain were merely studied. However, even if AD could be mitigated by removing the molecular accumulation of harmful proteins, we still need to revise the neural circuitry alterations caused by AD development to restore cognition and memory. Recently, several studies have been focusing on the circuitry alterations in AD brains[38,74,96,97], and the deep brain stimulation in the basal forebrain has become a promising approach for AD treatment[96,98]. However, the circuit mechanism of PFC dysfunction in AD brains remained unclear. In this study, we provided a circuit mechanism of how cholinergic inputs to ET neurons in the mPFC regulated ORM, which highlighted a potential target for the treatment of cognition impairment and PFC dysfunction caused by AD development. Finally, it is also noteworthy that all transgenic AD mouse models can be artificial in their genetic constitution. The overexpression of APP in the 5×FAD mouse model makes it even more artificial. It is unlikely that the rapid accumulation of Aβ in 5×FAD mice brain can fully mimic peptide-mediated effects in the human brain, in which the disease evolves over years and decades, and several pathologies develop in a time- and brain-region-dependent manner. Whether the circuit mechanism found in the present study can be applied to the treatment of patients and clinical research needs to be

further investigated. Another limitation of the present study is that we only explored the structural and functional alteration in female 5×FAD mice. The structural and functional alteration of neural circuits in male 5×FAD mice and why the female 5×FAD mice showed more severe symptoms are also worth exploring in future studies. In general, our study provides insights into the neural circuitry alterations in the 5×FAD mouse model, which may shed light on AD pathophysiology.

## Methods

**Animals**. The heterozygous 5×FAD mice line of C57B/6 J background express human APP and PSEN1 transgenes with a total of five FAD mutations: KM670/ 671NL (Swedish), I716V (Florida), V717I (London) mutations in APP, M146L (A > C), and L286V mutations in PSEN1 as previously described[37]. C57BL/6 J, ChAT-ires-Cre mice (stock No: 018957), and 5×FAD mice (stock No: 034848) were purchased from Jackson Laboratory (Bar Harbor, ME, USA). Fezf2-CreER mice in C57B/6 J background was a gift from the Josh Huang Lab. Ai47 reporter line was a gift from Hongkui Zeng of the Allen Institute for Brain Science. Heterozygous male 5×FAD mice were crossed with Fezf2-CreER females and ChAT-Cre mice, respectively, and the progeny were genotyped by PCR. The female 5×FAD mice, 5×FAD$_{Fezf2-CreER}$, and 5×FAD$_{ChAT-Cre}$ at two, five and six months of age and the same age of WT (C57, Fezf2-CreER or ChAT-Cre) were used for experimental studies. All animals were housed in a room under the conditions of 21 ± 1°C (humidity: 40–70%) and 12-hour light/dark cycle with lights on at 8:00 am with food and water ad libitum. Female animals were employed for all the animal experiments. All animal experiments were approved by the Animal Care and Use Committee of Huazhong University of Science and Technology.

**Viral vectors**. The AAV2/9-EF1α-DIO- GCaMP6s (2 × 10$^{12}$ gc/ml), AAV2/9-EF1α-DIO-NpHR-mCherry (2 × 10$^{12}$ gc/ml), AAV2/9-EF1α-DIO-ChR2-mCherry (2 × 10$^{12}$ gc/ml), AAV2/9-EF1α-DIO-GFP (2 × 10$^{12}$ gc/ml), AAV2/9-EF1α-DIO-mCherry (2 × 10$^{12}$ gc/ml), AAV2/9-EF1α-TVA-mCherry (2 × 10$^{12}$ gc/ml), AAV2/ 9-CAG-RG (2 × 10$^{12}$ gc/ml), RV-EnvA-ΔG-GFP (2 × 10$^{9}$ IFU/ml, used for RV tracing in 6-month animals), AAV2/9-EF1α-ACh2.0 (2 × 10$^{12}$ gc/ml), AAV-mCaMKIIa -ChR2-GFP(2 × 10$^{12}$ gc/ml), and AAV2/2Retro-Cre (2 × 10$^{12}$ gc/ml) were purchased from BrainVTA (BrainVTA Co., Ltd., Wuhan, China). AAV2/9-EF1α-DIO-hM4D (Gi) -mCherry (2×10$^{12}$ gc/ml), AAV2/2Retro-hSyn-GCaMp6s (2 × 10$^{12}$ gc/ml), AAV2/9-EF1α-fDIO-hM3D (Gq) -mCherry (2 × 10$^{12}$ gc/ml), and RV-EnvA-ΔG-GFP (1 × 10$^{8}$ IFU/ml, used for RV tracing in 2-month animals) were purchased from Taitool Bioscience (Taitool Bioscience Co. Ltd, Shanghai, China). The CAV-flex-Flp (5 × 10$^{12}$ gc/ml) was purchased from Montpellier vectorology. AAV-fDIO-TVA-GFP (2 × 10$^{12}$ gc/ml) was a gift from the Josh Huang Lab.

**Stereotaxic injections**. All mice were deeply anesthetized by intraperitoneally injected (100 g/ml) with 2% chloral hydrate and 10% urethane-configured anesthetic before they were mounted and microinjected with a stereotaxic system. For the fiber photometry in Figs. 1 and 6, 300 nL AAV-EF1α- GCaMP 6 s or the AAV-EF1α-ACh2.0 were injected into the PL (bregma 1.9 mm, lateral ±0.3 mm, depth 2.3 mm from skull surface) of both the 5×FAD$_{Fezf2-CreER}$ and Fezf2-CreER mice bilaterally at five months of age, respectively. The optical fibers (diameter, 200 μm, NA = 0.37, Newdoon Inc. China) were implanted 300 μm above the virus injection sites. Three days after the virus injection, tamoxifen (20 mg/ml) was injected intraperitoneally to induce the Cre recombination (1 ml/100 g body weight). The recording was performed three weeks after the virus injection.

For the optogenetics in supplementary Figure 7, 10, 300 nL AAV-EF1α-DIO-NpHR-mCherry was injected into the PL of Fezf2-CreER mice bilaterally at five months of age. Three days after the virus injection, tamoxifen (20 mg/ml) was injected intraperitoneally to induce the Cre recombination (1 ml/100 g body weight). To inhibit the somas and the specific axon terminals in different brain regions, the optical fibers (diameter, 200μm, NA = 0.37, Newdoon Inc. China) were implanted in PL (bregma 1.9 mm, lateral ±0.3 mm, depth 2.0 mm from skull surface), SI (bregma 0 mm, lateral ±1.25 mm, depth 4.5 mm from skull surface), SUM (bregma -2.7 mm, lateral ±0.7 mm, depth 4.7 mm from skull surface), and VTA (bregma -3.6 mm, lateral ±1.0 mm, depth 4.3 mm from skull surface). For the optogenetics in Figs. 2 and 3 and Supplementary Fig. 8, 300 nL AAV-EF1α-DIO-ChR2-mCherry was injected into the PL of 5×FAD$_{Fezf2-CreER}$ or Fezf2-CreER mice bilaterally at 5 months of age. Three days later, tamoxifen (20 mg/ml) was injected to induce the Cre recombination (1 ml/100 g body weight). For the soma and axon terminal activation, the optical fibers (diameter, 200 μm, NA = 0.37, Newdoon Inc. China) were implanted in PL, SI, SUM, and VTA. The behavior tests were performed three weeks after the virus injection.

To trace the whole brain axon projections of the Fezf2 positive neurons in the mPFC in Supplementary Figure 1, 200 nL AAV2/9-EF1α-DIO-GFP was injected into the PL of Fezf2-CreER mice at two months of age. Three days after the virus injection, tamoxifen (20 mg/ml) was injected intraperitoneally to induce the Cre recombination (1 ml/100 g body weight). For the Fezf2 positive neurons sparse labeling in mPFC, 150 nL AAV-fDIO-TVA-GFP was injected into the PL of Fezf2-CreER mice at two months of age. At the same time, 300 nL CAV-flex-Flp was

injected into the SI (bregma 0 mm, lateral 1.25 mm, depth 4.7 mm from skull surface) or SUM (bregma -2.7 mm, lateral 0.7 mm, depth 5 mm from skull surface). Three days after the virus injection, tamoxifen (20 mg/ml) was injected intraperitoneally to induce the Cre recombination (1 ml/100 g body weight). To label the cholinergic axon fibers in PL, 300nLAAV2/9-EF1α-DIO-GFP was injected into the horizontal diagonal band (bregma 0 mm, lateral ±1.25 mm, depth 5.3 mm from skull surface) of 5×FAD$_{ChAT-Cre}$ or ChAT-Cre mice at five months of age. Three weeks after the virus injection, the mice were killed and the brain tissues were processed for imaging. For the dual-color CTb labeling, 200nLCTb conjugated with Alexa 488 (Life Technologies) was injected into the SI and 200nLCTb conjugated with Alexa 594 (Life Technologies) was injected into the SUM. Seven days after the injection, the mice were killed and the brain tissues were processed for imaging.

For the light activation of cholinergic fibers in mPFC in Fig. 7 and supplementary figure 17, 300 nL AAV-EF1α-DIO-ChR2-mCherry was injected into the horizontal diagonal band of 5×FAD$_{ChAT-Cre}$ or ChAT-Cre mice bilaterally at five months of age. The optical fibers (diameter, 200 μm, NA = 0.37, Newdoon Inc. China) were implanted into the PL bilaterally to activate the cholinergic axon terminals in the PL. For the light inhibition of cholinergic fibers in mPFC in supplementary figure 16, 300 nL AAV-EF1α-DIO-NpHR-mCherry was injected into the horizontal diagonal band (bregma 0 mm, lateral ±1.25 mm, depth 5.3 mm from skull surface) of ChAT-Cre mice bilaterally at five months of age. The optical fibers (diameter, 200μm, NA = 0.37, Newdoon Inc. China) were implanted into the PL bilaterally to inhibit the cholinergic axon terminals in the PL. The behavior tests were performed three weeks after the virus injection.

For the activation of the ventral hippocampus of 5×FAD mice, 300nLAAV-camkii-ChR2-GFP was bilaterally injected into the ventral hippocampus (bregma -3.4 mm, lateral ±3.5 mm, depth 4 mm from skull surface) of 5×FAD mice at five months of age. The optical fibers (diameter, 200 μm, NA = 0.37, Newdoon Inc. China) were implanted into the ventral hippocampus bilaterally to activate the somas of the hippocampal neurons. The behavior tests were performed three weeks after the virus injection.

For the monosynaptic rabies tracing, 150 nLmixture of AAV-EF1α-TVA-mCherry and the AAV-CAG-RG were injected into the PL of 5×FAD$_{Fezf2-CreER}$ and Fezf2-CreER mice at 1.5 months or five months of age (The TVA and RG was 1:2). Three days after the virus injection, tamoxifen was injected to induce the Cre recombination. Three weeks later, the RV-EnvA-DG-GFP was injected into the same area. One week after the RV injection, the mice were killed and the brain tissues were processed for the imaging. To test the specificity of the helper virus, AAV-EF1α-TVA-mCherry was injected into the vertical diagonal band (bregma 0.7 mm, lateral 0 mm, depth 4.7 mm from skull surface) of ChAT-Cre mice. In the meanwhile, RV-EnvA-DG-GFP was injected into the PL. One week after the virus injection, the mice were killed and the brain tissues were processed for imaging.

For the chemogenetical manipulation and fiber photometry in Fig. 8 and supplementary figure 18, 300 nL AAV2/9-EF1α-fDIO-hM3D(Gq)-mCherry was injected into the horizontal diagonal band of 5×FAD$_{ChAT-Cre}$ mice bilaterally. In the meanwhile, 300nL CAV-flex-Flp was injected into the PL bilaterally to provide the Flp in the horizontal diagonal band for hM3D(Gq) expression. 300 nL AAV2/2Retro-hSyn-GCaMp6s was injected into the SUM bilaterally to label the PFC-SUM projection neurons for fiber photometry. The optical fiber was implanted into the PL for recording. For the chemogenetics inhibition experiments, 300 nL AAV2/9-EF1α-DIO-hM4D (Gi)-mCherry was injected into the horizontal diagonal band of ChAT-Cre mice bilaterally to inhibit the activity of cholinergic neurons in the horizontal diagonal band. 300 nL AAV2/2Retro-hSyn-GCaMP6s was injected into the SUM bilaterally to label the PFC-SUM projection neurons for fiber photometry. The optical fiber was implanted into the PL for recording. For the C57BL/6 and AD control group, AAV2/2Retro-hSyn-GCaMP 6s is injected into the SUM bilaterally to label the PFC-SUM projection neurons for fiber photometry. The optical fiber was implanted into the PL for recording. One weeks after the virus injection, the mice which expressed hM3D(Gq) or hM4D (Gi) were provided with drinking water containing CNO (ApexBio, 5 mg/kg/day) and saccharin (5 mM, Sigma). The CNO administration lasted one week, and then, the fiber photometry was performed. Next, the drinking water containing CNO was replaced with ordinary drinking water. One day after the replacement, the fiber photometry was performed again. For the CNO control experiment, two groups of six-month old 5×FAD mice were given either drinking water containing CNO (ApexBio, 5 mg/kg/day) and saccharin (5 mM, Sigma) or ordinary drink water with saccharin (5 mM, Sigma) for a week. Then, the animals were assigned to object recognition test.

All virus injections were performed using a pulled glass micropipette at a speed of 60 nL min$^{-1}$ and delivered with a micro-syringe pump (Nanoject II, Drummond Scientific). Following the completion of viral injection, the needle was held for 10 min at the site and then retreated slowly. After that, incisions were stitched, and lincomycin hydrochloride and lidocaine hydrochloride gel was applied to prevent inflammation and alleviate pain for the animals. For the mice which were tested for fiber photometry or optogenetics, the dental cement and skull screws were applied to fix the optical fibers for further experiments.

**Histology and Immunohistochemistry.** Mice were intraperitoneally injected (100 g/ml) with sodium pentobarbital (1% wt/vol) and then perfused with 0.01 M PBS (Sigma-Aldrich) for 10 min, followed by 4% PFA (Sigma-Aldrich) for 10 min.

The mouse brains were removed and post-fixed in 4% PFA solution at 4 °C for 12 h. After that, the samples were rinsed with 0.01 M PBS for 12 h. For the immunohistochemistry, the mouse brains were sectioned at 70-μm thickness by a vibratome (Leica, VS1200S). The coronal brain sections were rinsed with 0.01 M PBS for 3 × 10 min and blocked with 5% (wt/vol) BSA in 0.01 M PBS (at 37 °C for 2 h). Next, the brain sections were incubated with the following primary antibodies (at 4 °C for 12 h): anti-c-fos (1:800, rabbit, Cell Signaling Technology, Cat# 2250, RRID: AB_2247211), anti-Aβ (1:800, mouse, abcam, ab126649), anti-chat (1:200, goat, Millipore, AB144P), and anti-NeuN (1:800, Rabbit, Abcam, ab7349). Following the incubation with the primary antibodies, the sections were rinsed with 0.01 M PBS for 3 × 10 min and then incubated with the following fluorophore-conjugated secondary antibodies (1:500, at 37 °C for 2 h): Alexa Fluor-594, donkey antigoat; Alexa Fluor-405, goat antimouse; Alexa Fluor-594 or 647, donkey anti-rabbit. Following this procedure, the brain sections were rinsed with 0.01 M PBS for 3 × 10 min. Finally, the brain sections were attached to glass slides and imaged with a commercial confocal microscope (Carl Zeiss, LSM710) or a slide scanner (Olympus VS120). For the rapid staining of Aβ, the brain sections were rinsed with 0.01 M PBS for 3 × 10 min and incubated with 1 μM of DANIR-8c (5% DMSO, 10% alcohol, 85% 0.1 M PBS) for 10 min at room temperature and then rinsed with 70% alcohol for 5 min.

The fluorescent in situ hybridization experiment to detect Cre expression in the mPFC was performed according to previous studies[99,100]. Briefly speaking, the 2-month old Fezf2-CreER or 2-month old 5×FAD$_{Fezf2-CreER}$ mice were perfused with PBS and PFA. The brain samples were collected and dehydrated with 30%(w/v) sucrose solution for three days. Then, the samples were sectioned at 12 μm, and the sections which contained the mPFC were collected for staining. The fixed sections were rapidly dehydrated through a 70%, 85%, and 100% graded pre-cold ethanol series. The nuclei were prehybridized with 350 μL of probe hybridization buffer (50% formamide, 5× SSC, 9 mM citric acid (pH 6.0), 0.1% Tween 20, heparin (50 μg/ml), 1× Denhardt's solution, and 10% dextran sulfate) for 30 min at 45 °C. After removing the prehybridization solution, the probe solution (500 μL of probe hybridization buffer containing 1 pmol of each probe. For the probe sequences, see Supplementary Table 1) was added and incubated for 16 h at 45 °C. The sequences of the probes to detect the Cre expression were listed in Supplementary Table 1. The hairpin solution was prepared by adding all the snap-cooled hairpins (30 pmol of each fluorescently labeled hairpin in 10 μL of 5× SSC buffer, heated at 95 °C for 90 s and then reduced stepwise to 25 °C at a rate of 0.1 °C/s) to 500 μL of amplification buffer (5 × SSC, 0.1% Tween 20, and 10% dextran sulfate) at room temperature. The sections were pre-amplified with 350 μL of amplification buffer for 30 min at room temperature. After removing the preamplification solution, the hairpin solution was added. Last, the slides were incubated overnight at room temperature. Slides were washed five times with 5× SSCT (5× SSC and 0.1% Tween 20). After washing, slides were stained with 4',6-diamidino-2-phenylindole (DAPI; Life Technologies) and analyzed under a fluorescence microscope.

For resin embedding, mice were deeply anesthetized with sodium pentobarbital (1% wt/vol) and subsequently intracardially perfused with 0.01 M PBS (Sigma-Aldrich), followed by 4% paraformaldehyde (Sigma-Aldrich) and 2.5% sucrose in 0.01 M PBS. The brains were excised and post-fixed in 4% paraformaldehyde at 4 °C for 12 h. For whole-brain imaging, the intact brains were embedded in glycol methacrylate (GMA) resin. Each intact brain was rinsed overnight at 4 °C in a 0.01 M PBS solution and subsequently dehydrated in a graded ethanol series (50, 70, and 95% ethanol, changing from one concentration to the next every 1 h at 4 °C). After dehydration, the brains were immersed in a graded GMA series (Ted Pella Inc.), including 0.2% SBB (Sudan black B) (70, 85 and 100% GMA for 2 h each and 100% GMA overnight at 4 °C). Subsequently, the samples were impregnated in a prepolymerization GMA solution for 3 d at 4 °C and embedded in a vacuum oven at 48 °C for 24 h. Each 100 g of GMA solution (100%) consisted of two resin components (A component, 67 g; B component, 29.4 g), 2.8 g of deionized water, 0.2 g of SBB, and 0.6 g of AIBN (2,2'-azo-bis-butyronitrile) as an initiator. The 70% and 85% GMA solutions (wt/wt) were prepared from 95% ethanol and 100% GMA.

**Microscopy.** For immunohistochemistry imaging, the sections were mounted with 50% glycerol (vol/vol) and imaged using a 20×, 0.75 NA objective (Zeiss 710) or a 10×, 0.4NA objective (Olympus VS120). For c-fos counting, all brain sections containing the brain areas of interest were obtained from mouse brains. Every second brain sections were collected, stained with c-fos antibody and mounted with 50% glycerol. Then, the brain slices were scanned using a 20×, 0.75 NA objective (Zen 2011, Zeiss 710) or a 10×, 0.4NA objective (Olympus VS120) (For example, in extended Fig. 5, the WT NOR group and the AD control group were imaged with Leica sp8 confocal microscopy; the WT control group and the AD NOR group were imaged with Olympus VS120 slider scanner). The cell counting was performed manually with ImageJ. For whole-brain precise imaging, the GMA embedded mouse brains were imaged by our homemade fMOST system. The imaging system has been described previously[23]. Briefly, the system used a mercury lamp (X-Cite exacte, Lumen Dynamics) as light source, a digital micro-mirror device (DMD, XD-ED01N, X-digit) to generate the illumination grid pattern and a water immersion objective (1.0 NA, XLUMPLFLN 20XW, Olympus) for imaging. Two scientific complementary metal-oxide-semiconductor cameras (ORCA-Flash

4.0, Hamamatsu Photonics K.K.) were used for signal detection. A piezoelectric translational stage (P-725 PIFOC Long-Travel Objective Scanner, E-753 Digital Piezo Controller, PI GmbH) moved the objective for axial scanning. The sample box was screwed onto a high-precision 3D translation stage (ABL20020-ANT130-AVL125, Aerotech Inc.). The 3D translation stage moved the sample for mosaic scanning and sectioning. A diamond knife (Diatome AG) was used for sample sectioning. During imaging, the sample was immersed in a water bath containing PI (1 µg ml$^{-1}$, wt/vol) and 0.05 M Na$_2$CO$_3$. The objective scanned the surface of the sample in mosaic mode at a step of 2 µm. After one surface was finished, the diamond knife removed the imaged surface and exposed the smooth fresh surface for imaging. The mosaic imaging process was repeated until the entire coronal section was acquired. After data acquisition, the images were preprocessed to generate a series of coronal images. The datasets for complete neuron morphology reconstructions were converted to other format for further process (Please see Visualization and reconstruction part). For whole-brain rabies virus labeled neuron counting, the coronal images were used to generate 50 µm maximum projection images. Then, the labeled neurons were manually counted with ImageJ according to the Allen brain atlas[101].

**Image preprocessing**. The raw data acquired by the brain positioning system needed image preprocessing for mosaic stitching and illumination correction. This process has been described before[23]. Briefly, the mosaics of each coronal section were stitched to obtain an entire section based on accurate spatial orientation and adjacent overlap. Lateral illumination correction was performed section by section. Image preprocessing was implemented in C++ and optimized in parallel using the Intel MPI Library (v.3.2.2.006, Intel). The whole data set were executed on a computing server (72 cores, 2 GHz per core) within 6 h.

**Visualization and reconstruction**. We visualized the data set using Amira software (v.5.2.2, FEI) to generate the figures and videos. The data set acquired by the dual-color precise imaging system was separated into the GFP channel and PI channel. The PI-labeled data set was sampled to $3.2 \times 3.2 \times 50$ µm$^3$ and imported into Amira to generate the outline of the mouse brain. To trace the morphology of the input neurons, we transformed the data format of GFP-labeled data from TIFF to TData via the algorithm developed by our lab[102]. A homemade software Gtree was used to trace the morphology of GFP-labeled neurons at the whole-brain level by human-machine interactions[103]. Briefly, we loaded the data block of interests into Amira and assigned the initial and terminal points of the fibers in the block, so that Amira could automatically calculate the pathway between initial and terminal points. We repeated this procedure until the reconstruction was finished. The reconstructed neurons were checked back-to-back by three people. The tracing results were saved in SWC format. We loaded the outline of the mouse brain and the tracing results into Amira simultaneously and used the moviemaker module of Amira to generate figures and videos.

**Behavioral assays**. The female mice at two months and six months of age were used to conduct behavioral experiments between 8:00 AM and 6:00 PM. For all behavior tests, all the experimental mice were transferred to the behavior testing room 3 days prior to beginning of the first trial so that they could habituate to the condition of the behavior testing room. The behavior chamber was cleaned with a 75% ethanol prior and after each test to remove any scent clues left by the previous subject mice.

**Open field test**. Mice were individually placed facing one of the walls of a Plexiglass open-field ($40 \times 40 \times 30$ cm). Mice were allowed to explore freely for 10 min, and the time spent in the center zone and the outer zone were automatically tracked using EthoVision XT 12.0 (Noldus Apparatus).

**Novel object recognition test (NOR)**. The NOR test was similar to that in the previous study[27]. The test includes three sessions of one trial each: training (memory acquisition), delay (memory consolidation), and testing (memory expression) trials. For three consecutive days, mice were individually habituated to the open field for 10 min. For the training trial, mice were placed in an arena that contained two identical objects for 10 min. The mice that did not explore the objects for 10 s within the 10 min period were excluded from the further experiments. A mouse was scored as approaching with the object when its nose within 2 cm of the object. Then, the tested mice were returned to the home cage for memory consolidation (delay). The testing session was done 10 min after the acquisition trial. In this trial, one of the objects presented in the first trial was replaced with a novel object. Then, the tested mice were placed back in the arena for 10 min, and the total time spent in the exploration of each object was recorded. Time spent in active exploration of the familiar (F) and novel (N) objects during the retrieval trial were calculated using EthoVision XT 12.0 (Noldus Apparatus). Recognition memory was scored using a recognition index for each mouse with a formula (N or F) / (N + F) %. The recognition index reflected the difference between the time exploring the novel and familiar objects and the total time exploring both objects. The delta recognition index was calculated by the recognition index of novel object minus the recognition index of familiar object. Normally, the test mice will spend more time with the novel object. The preference

towards the novel object indicates the ORM of the tested mice. The delta recognition index = recognition index(N) - recognition index(F). For the c-fos staining, each tested mouse was kept separately in individual homecage 24 h before the NOR. Then, each tested mouse was subjected to NOR through three phases. One and half hours after the test, the tested mouse was intraperitoneally injected (100 g/ml) with sodium pentobarbital (1% wt/vol), then perfused with 0.01 M PBS (Sigma-Aldrich) for 10 min, and followed by 4% PFA (Sigma-Aldrich) perfusion for 10 min. The brain was removed for c-fos immunochemical staining.

*iClass test*. The iClass test was similar to the previous studies[104,105]. The test mice that expressed ChR2 or NpHR or GFP were individually habituated to the open field for 10 min for three consecutive days. Then, the test mice were tested in the pre-session in which no light was delivered during the test. The traveling distance and time in the center area of the arena and the outer area of the arena (the size of the arena, 40 cm×40 cm×30 cm; center area, 20 cm×20 cm) was recorded by EthoVision XT 12.0 (Noldus Apparatus). The next day, the test mice were tested in the L1 session in which light stimulation was delivered whenever the centroid of the mouse body was located within the center area of the arena. The test lasted for 10 min. The traveling distance and time in the center area of the arena and the outer area of the arena were recorded by EthoVision XT 12.0 (Noldus Apparatus). Then, the next day the test mice were tested again in L2 session with the same approach used in L1 session. The classical iClass test often contained a post/extinction session. Since we found that in our results, manipulation of ET neurons in the mPFC did not induce the robust conditional place preference or aversion. Therefore, we skipped the post/extinction session. The parameter of light stimulation: blue light activation, 3.5–5 mW at the tip of the optical fiber, 10 Hz, 15 ms pulse width; yellow light inhibition, 3.5–5 mW at the tip of the optical fiber, insistent inhibition.

**Optogenetics**. For manipulation of somas and axon terminals of the Fezf2 positive neurons in mPFC, after the virus injection, the optical fiber was planted into the PL or other brain areas (SI, SUM and VTA) of the tested mice. The ceramic ferrule was supported with screws and dental cement to the skull. A month after the virus injection, the optogenetics manipulations were performed. The mice were coupled to a single-channel fiber optic patch cord (diameter, 200 µm, 0.37 NA) that connected to a 473-nm laser or 570-nm laser for optogenetic stimulation. The activation experiments were conducted with 15 ms pulses width of 473-nm light at 10 Hz, and photoinhibition experiments were conducted with persistent inhibition. The stimulation parameters were determined according to our previous study as well as other similar studies[50,51,54]. The stimulation frequency was slightly modified since we found that 20 Hz of laser activation of somas of the Fezf2 positive neurons in the mPFC can cause seizure.

To explore the role of Fezf2 positive neurons or the cholinergic neurons in the different phases of object recognition test, the laser was delivered at different phase of object recognition test for 10 min to cover the whole duration of that phase. The laser power of both activation experiments and inhibition experiments was about 3.5–5 mW at the tip of the optical fiber which connected with the mouse brain. To test the Fezf2 positive neuronal activity is crucial for recognizing the familiar objects, we also conducted experiments to inhibit the Fezf2 positive neuronal activity only when the animals entered the zone where the familiar or novel objects were located. The inhibition laser was turned off when the animals left the object location. For cholinergic fiber manipulation experiments in PL, after the virus injection, the optical fiber was planted into the PL of 5×FAD$_{ChAT-Cre}$ and ChAT-Cre mice at five months of age. The manipulation parameters were similar to those in the manipulation of somas and axon terminals of the Fezf2 positive neurons in mPFC. During all the behavior tests coupled with optogenetics, the tested mice were first tested without light stimulation via NOR to access their cognition level. Then, the mice were returned to homecage to rest and tested again on the next day with light stimulation to evaluate the impact of the light manipulation. Between two tests, different sets of objects were used during NOR to prevent memory generalization.

**Fiber photometry**. Calcium signal and the acetylcholine release was recorded by a commercialized fiber photometry system (Thinker Tech Nanjing Biotech CO., Ltd). After the virus injection, the optical fiber (diameter, 200 µm, NA = 0.37) was planted into the PL of both Fezf2-CreER mice and AD$_{Fezf2-CreER}$ mice at five months of age. The ceramic ferrule was supported with screws and dental cement to the skull. Three weeks after virus injection, monitoring the activity of ET projection neurons and the acetylcholine release in the mPFC were performed, respectively. Fiber recordings were performed in freely moving mice during the object recognition test. To induce fluorescence signals, a laser beam from a laser tube (488 nm) was reflected by a dichroic mirror, focused by a 10× len (NA = 0.3) and then coupled to an optical commutator. A 3-m optical fiber (200 mm O.D., NA = 0.37) guided the light between the commutator and the implanted optical fiber. To minimize photobleaching, the power intensity at the fiber tip was adjusted to 0.02 mW. The GCaMp6s and the acetylcholine indicator fluorescence was bandpass filtered (MF525-39, Thorlabs) and collected by a photomultiplier tube (R3896, Hamamatsu). An amplifier (C7319, Hamamatsu) was used to convert the photomultiplier tube current output to voltage signals, which was further filtered through a low-pass filter (40 Hz cut-off; Brownlee 440). The analog voltage signals were digitalized at 100 Hz and recorded by a Power 1401 digitizer and Spike2 software

(CED, Cambridge, UK). The calcium signals and the object interactions were simultaneously recorded by data acquisition software (Thinkertech, China), and the calcium data from each continuous experimental trial was normalized to the averaged fluorescence by a MATLAB program developed by Thinkertech. Briefly, raw signals were first adjusted according to the overall trend to account for photo-bleaching before further analysis. For each trial, the averaged baseline signals before stimuli presentation were considered as "F0" and fluorescence change (DF/F) was calculated as (F-F0)/F0 unless otherwise stated. The object interactions were defined as the time point when the mice poked the object or sniff the object with their noses. The time points of object interactions were automatically recorded by calcium acquisition software (Thinkertech, China) or EthoVision XT 12.0 (Noldus Apparatus) and manually checked to assure the precision of the recording. For alignment of calcium signals and object interactions, the time point of object interactions was transferred into binary form (0 or 1) and for any behavior, and any time points with a specified behavior annotated were "1" and otherwise as "0". DF/F values were segmented based on time point of interactions and averaged first across different trials and then across animals. For the quantification of calcium signals during interactions with old or novel object, DF/F values were first averaged for each trial for the time points that were annotated with a certain behavior and then averaged across trials and animals. For this calculation, the baseline was taken as the mean signal between -2s to -1s (1-2 s before the interaction begins, relative to behavior onset "0"). We calculated the average delta F/F during the object interaction. The non-parametric Wilcoxon signed-rank test or $t$ test was used to determine whether the AUC during the interaction time window were significantly different from control (interaction time window 2 s before and 4 s after the object interaction). The interaction time window was chosen according to previous studies which conducted similar analysis[51,106,107].

**Sample sizes statement.** No statistical methods were used to predetermine sample size; however, the sample size was similar to previous studies[27,104,105].

**Replication.** Results described throughout the paper were reproduced. 5-10 rounds of experimentation were performed in independent animals. The n value in the paper indicated the number of replications. No result was included that were not observed in multiple animals. No issue was identified in replicating any of the reported findings.

**Randomization.** For all experiments in this study, the animals were randomly assigned to experimental groups and control groups. For the comparison between AD group and wild type group, the heterozygous 5×FAD mice were randomly assigned to the AD group, and the noncarriers of AD mutant genes from same genetic background were randomly assigned to the wild type group.

**Blinding.** For the behavior experiments including fiber photometry, optogenetics and chemogenetics, the investigator was blinded to the groups. For rabies tracing, the person who analyzed the data was binded to the strain of the animals. For other tracing experiments such Supplementary Fig. 1, the person who performed the experiments was not blinded to the strain of the animals since no comparison was required in those experiments.

**Statistical Analysis.** Statistical significance was analyzed using GraphPad Prism version 6.0. The average plots across different trails of the individual animals were calculated with MATLAB. The AUC of the average plots was also calculated using GraphPad Prism version 6.0. All measurements were listed as Mean ± SEM. Statistical comparisons were performed using Student's $t$ test and ANOVA. Statistical significance was defined as $p < 0.05$.

**Reporting Summary.** Further information on research design is available in the Nature Research Reporting Summary linked to this article.

## Data availability

All the data that support the findings of this study are provided in the article and its Supplementary information files and source data. Raw data for neuron reconstruction is available at http://atlas.brainsmatics.org/a/sun2112. Source data are provided with this paper. Additional information about this paper are available from the corresponding author upon reasonable request. Source data are provided with this paper.

## Code availability

The code for image processing is available in previous studies[23,108], which can be found from http://atlas.brainsmatics.org/a/zhong2019. The code for fiber photometry is available from http://atlas.brainsmatics.org/a/sun2112.

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

## Acknowledgements

We thank Josh Huang of Duke University for providing the Fezf2-creER mice and critical comment on the manuscript. We thank Tao Jiang, Xueyan Jia, and members of HUST-Suzhou Institute for Brainsmatics for help with experiments and data analysis. We thank Yulong Li from Peking University for providing the acetylcholine indicator. We thank Xiaojun Wang from Hainan University for the helpful comments on experiments and manuscript. We thank the Optical Bioimaging Core Facility of HUST for support with data acquisition. This work was financially supported by the National Natural Science Foundation of China (Nos. 91749209, 61890953, 31871088), CAMS Innovation Fund for Medical Sciences (2019-I2M-5-014) and the Director Fund of WNLO.

## Author contributions

Q.L., H.G., and X.L. conceived and designed the study. Q.S. and J. Z. performed most of the experiments and analyzed the data. M.Y. helped with the virus injection and opto-genetic experiments. S.C. performed the brain sample processing. G.L. and Z.W. performed the whole-brain data acquisition. A.L. and Y.L. performed the imaging processing and neuron reconstruction. Q.S., X.L., H.G., and Q.L. wrote the paper with inputs from other authors.

## Competing interests

The authors declare no competing interests.
