## [Peer Review File · Nature Communications]

Reviewers' Comments:

Reviewer #1:

Remarks to the Author:

The manuscript by Sun et al explores the role of cholinergic modulation of layer 5 mPFC neurons in novelty encoding in a mouse model of Alzheimer's disease. The manuscript itself showcases a number of powerful techniques that reveal some interesting findings with respect to the connectivity of layer 5 neurons and functional specificity of mPFC terminals in the supramammillary nucleus. Enthusiasm for the work is mixed, however. Important considerations of the animal model, and the neurobiological impact of the chemogenetic and optogenetic effects that were used need further discussion and or validation (more below). There are also several points that need to be addressed with respect to the experimental design and statistical analysis that limit my ability to evaluate of the robustness of the reported results. General and specific points listed below.

General comments

1. The manuscript focuses on the role of a subset of neurons in the mPFC and novel object recognition, although previous work has repeatedly shown that full lesions of the mPFC have no effect on novel object recognition (Ennacuer et al, 1997; Barker et al., 2007; Cross et al., 2013). This raises the question of circuit specificity and neurobiological relevance of the techniques used to manipulate the PFC as described in the study (more below).
2. With respect to the 5XFAD mice in particular, previous data has suggested that by 6 months of age significant plaque accumulation and signs of neurotoxicity are observed in both the hippocampus and layer 5 of the frontal cortex (Oakley et al. 2006; Eimer et al. 2013) Given the very clear role of the hippocampus in novel object recognition, especially in ORT with a short consolidation period as in the present study, as well as the role of SUM projections to DG in memory retrieval (Li et al., 2020), it is important to understand what PFC dysfunction is doing that is unique from the behavioral effects of hippocampal dysfunction. Sex as a biological variable should also be addressed, particularly given that the study was only run in female mice which appear to have greater amyloid deposition in this model. It would also be nice to know how consistent AB accumulation was between mice, and whether or not that correlated with any behavioral effects.
3. The manuscript would benefit greatly from acknowledging more of the literature on this circuitry that has accumulated over the decades. In particular, I would suggest the authors take time to explore work by Martin Sarter, Mike Hasselmo, Huib Mansvelder, and Evelyn Lambe for additional information on what is already known about cholinergic modulation of deep layers of the mPFC, how this relates to cognitive function, and the impact of chronically increasing or decreasing ACh release in the mPFC with DREADDS relative to how the system is normally engaged.

Specific strengths

1. The power of the mPFC terminal manipulation in the SUM is really quite compelling. Again, understanding if this effect was unique to mPFCSUM interactions, or, if it reflected an indirect on SUMHippocampus modulation of memory would be very interesting.
2. Throughout, the anatomical description and data are very interesting and well done.

Issues remaining

1. Throughout the manuscript, it needs to be made clear what statistical comparisons were made. The F/T values, DF, and exact p-values should be included. It is also not clear that ANOVAs or T-tests were necessarily the best test of differences in central tendency for these groups. For example for photometry data, were signals mouse or trial weighted? There is a plot of a test on AUC, but what about event amplitude, event frequency. Why were these tests selected? Why these time windows? Given the effects of optogenetic stimulation on locomotor output in several conditions, a regression controlling for the effect of locomotor output on target-associated Ca²⁺

dynamics would be informative.

2. Why was a 10 minute delay used for the ORT experiments? Were longer delays tested?

3. Did the AD-Fezf2-CreER cross show a quantitatively similar pattern of Cre expression in the mPFC?

4. With respect to the photometry data in Figure 1, more information is needed. Later in the manuscript it is reported that Layer V neurons exhibit a greater response to familiar objects relative to novel. Is the reduced amplitude of evoked mPFC responses in the AD mice specific to the familiar object? Is activity overall reduced? Did response magnitude to the familiar object correlate with overall performance in this cohort?

5. For the c-fos/GFP examination depicted in Figure 1 M-P, what exactly was the N for each of the groups? Was there a control condition for the AD mice as well? Although the author's state there is "no obvious difference" in the number of GFP cells between AD and non-AD mice, the representative images suggest there are indeed fewer cells in the AD mice. It would be important to rule this out with a larger cohort and fewer neuron number could easily account for the difference in the photometry signal with no need to invoke the cholinergic system.

6. For the data described in Supplemental Figure 3, was the total cell count (not just fos+) reduced in the AD model? In other words, did the PFC not recruit the structure, or, was there less of the structure to recruit?

7. For the optogenetics experiment in Figure 2, as well as the rest of them (e.g Figure 3, 7, Supplemental 10), how was specificity of opsin expression confirmed? How were stimulation parameters determined? When exactly is the laser being turned on? Why at that time? If augmented mPFC responses to familiar stimuli are what is important for ORT, were experiments with opto used to test this hypothesis directly? In some places it is noted that optical stimulation/inhibition affected locomotor output, but this is not addressed for every opto experiment and really should be to facilitate interpretation of the effects, particularly the impaired ORT in WT mice following optogenetic stimulation and inhibition.

8. For Figure 5, seems surprising and potentially important if indeed there is no input from nucleus basalis (NBm) to layer 5 PFC. Some quantification here would be nice. The GFP-based anterograde tracing would also be strengthened by a quantification of AChE fiber density in PFC.

9. For the DREADD experiments, a CNO control group is necessary to control for the effects of a week of being exposed to low dose, back converted into clozapine. Also, important to confirm that in animals that exhibited behavioral effects, virus expression was indeed confined to the horizontal band and did not spread to nearby cholinergic cell groups that project to other potentially important brain regions.

Reviewer #2:

Remarks to the Author:

In their article, Sun and colleagues provide evidence that mPFC circuit impairment underlies decreased object recognition memory in a mouse model of Alzheimer's disease. The authors use state-of-the-art opto- and chemogenetic tools combined with in vivo behaviour to trace down an impaired neurocircuit in the 5xFAD mouse model. The paper thereby presents a novel brain circuit (ACh neurons BF  ET neurons in mPFC cortex  Undefined neurons in SUM) that is engaged in healthy subjects during Object Recognition, and is impaired in AD.

Overall, the study is carefully conducted with a high quality of the used methods and concisely described in appealing figure designs. Convincingly, the authors trace and describe a neurocircuit underlying object recognition memory in mice and pinpoint degeneration of cholinergic neurons as

a hallmark in circuit dysfunction associated with memory loss. While it is not new that cholinergic neurons undergo degeneration in humans and various mouse models of Alzheimer's disease, the present work elegantly depicts the consequences on neurocircuits. (Over)activation of the remaining cholinergic system, or activation of second/third order neuronal populations however is sufficient to restore object recognition memory in the used AD model. Importantly, the authors transfer the progress of the last decade in systems neuroscience methods to a disease relevant model and highlight the opportunities using these methods outside of basic research to analyze disease progression and clinically relevant disease model in more detail.

Major:

1. Mouse model. The authors choose the 5xFAD mouse as a model system to analyze functional impairment of basal forebrain cholinergic – mPFC circuits to underly cognitive deficits in Alzheimer's disease. However, neither in the introduction nor in the discussion, the authors clarify why exactly this AD model has been chosen. Given the variety of different AD mouse models, the authors should clearly describe the rationale behind their decision and describe advantageous and possible disadvantageous of the model. The rationale should already be given in the introduction while the rest should be included in the discussion in more detail also comparing their results with other models and integrate into the literature.

2. Development of cognitive dysfunction and disease progression. The authors provide convincing behavioral data at two different time points (2 and 6 month) upon which they decide to further perform their experiments in the older animals. While these experiments are a well-chosen basis of the whole project, enthusiasm could be further increased if the authors would provide a more precise timeline of disease progression. While performing experiments again at multiple timepoints certainly is beyond the scope of reasonable request, a detailed immunohistochemical analysis in combination with further retrograde or anterograde tracing could provide important information. More precisely:

No behavioral phenotype is detected in 2-month old animals. When is the first decline in PFC-related memory impairment detectable? The 5xFAD mouse model shows plaques and gliosis already at the age of 2 month. For future studies in the same and other groups this would provide important information when to further expand on the interesting and important findings reported in the present manuscript

a) Does the memory decline coincide with cholinergic neuron loss in the basal forebrain and/or fiber loss in the mPFC?

b) Does fiber loss in the mPFC precede neuronal loss in the basal forebrain?

To address these points, we suggest the authors add in a two month time point with methods utilized in figure 5 (a-c), and provide a quantification of axonal integrity (d-g) for the already acquired 6 month time point and a 2 month timepoint. The nicely designed experiment indicates terminal loss precedes soma loss, as discussed by the authors. However we feel that given the presented variability in soma counts across individual mice at 6 months, Inclusion of the 2-month time point and quantification of axonal integrity at 2 months and 6 months would greatly benefit the community and the impact of the presented work.

3. The title of the paper implies that circuit impairment resides at the site of projection neurons of the mPFC, however, the authors elegantly identify basal forebrain cholinergic neuron soma and fiber loss as the main driver of circuit dysfunction. Hence the title of the paper should be slightly adjusted accordingly, in order to better disseminate message/target relevant audience.

Minor:

1. Please clarify whether heterozygous/homozygous 5xFAD mice were employed in the study. If mixed groups were used, can the authors find any significant differences in ORT performance as a function of zygosity? Can this factor help explain phenotypic variance (ie, spread of data)

2. For clarity, please refer to training/delay/testing with consistent phrases in main text and all

other sections, (sometimes delay is referred to as 'consolidation', sometimes testing referred to as 'expression phase')

3. Were differences in soma counts found in the rabies retrograde tracing experiments in Fez-cre mice? Statement in text is ambiguous and not further explained 'We found that the proportion in most of the input brain areas were similar between 5xFAD and WT mice'. What precise presynaptic areas were affected? Was this limited to the basal forebrain as further described? What was the mean number of soma counted? Plots currently report proportion of total (%)

4. Further typos in text should be revised. Also typos in axis label across should be checked for (for example figure1e, fig1k)

Reviewer #3:

Remarks to the Author:

This is an interesting study investigating the neural circuitry underlying impaired novel object recognition in Abeta plaque-bearing 5xFAD mice. The study provides novel mechanistic insight into the loss of novel object preference in this mouse model by showing that decreased cholinergic input to ET neurons in the mPFC appears to be the major driver of this deficit. However, several technical, analytical and conceptual issues reduce my enthusiasm for the study, and the relevance of their findings for the human condition remain uncertain.

The study is motivated by 'clinical and animal model studies which demonstrated impairment of object recognition memory in Alzheimer's disease (AD)' and seeks to examine 'the underlying neural circuit mechanism'. However, to achieve this the authors need to be much more precise (e.g, by using adequate references) at which stage of the disease such deficit occurs, and this then should inform the choice of mouse model. If such deficits occur in individuals with AD dementia then using an Abeta model alone may not be sufficient to model this stage, and what we learn from the mice may not be relevant to human disease. Alternatively, they could remove all claims of human relevance and simply state that they are investigating the circuitry that underlies novel object recognition memory deficit in this particular mouse model (also keeping in mind that other models may not show deficits or deficits before plaques emerge, and the mechanisms may be different).

Information on statistical analysis is inadequate in my opinion, and as far as I can ascertain the relevant information is not present in the results section, figure legends or methods which makes it impossible to assess the validity of some of the results. They list statistical comparisons they made in one of the excel files (303849_0_source_data_891366) but it doesn't contain all the info they claim to have provided in the reporting table (eg, normality testing)?

5xFAD are available both on mixed background and congenic BL/6J background and it is important to know which they used for their cross-breedings as they may end up comparing mice on different background strains which is a confound. Can they specifically state the backgrounds of their mice and that all genotypes used are on the same background? Also, it appears that they are only using female mice, can they please clarify and also provide the rationale for this?

The focus is on layer 5 mPFC in 5xFAD mice but I would like to see as a negative control and to demonstrate specificity that the same deficits are not present in another brain region, in particular hippocampus which is thought to be involved in novel object recognition.

Fig 1, calcium data: the data suggest that the amplitude of population calcium transients in 5xFAD is reduced compared to WT. Amplitude difference may also be explained by fewer cells expressing GCaMP in 5xFAD and other technical confounds, which needs to be clarified.

Fig 1, cfos data: There is almost no background in 'AD NOR' compared to 'WT NOR'. C-fos antibody can also stain plaques. My first thought was that given this, the higher background in the WT-NOR could be just because more excess antibody was sequestered by the plaques in the 5xFAD group. But actually, the dark cross in the WT-NOR suggests over-exposure and tiling effects, and the GCaMP in that group also looks odd compared to the others. C-fos quantification can be especially sensitive to brightness/background (and may be masked by the plaques), so I think this is an

issue that needs clarification as it may impact on their results.

Figs 2d, e: Is this the same mouse shown here? Please clarify.

Fig 2g: It appears that the negative control for ChR2 in ADFezf2-CreER is expression of GFP but I would also like to see the effects of ChR2 expression in Fezf2-CreER.

Fig 6i: 'on multiple animals', please clarify number of animals used. This is a general issue throughout that information on number of animals and statistical tests is missing.

Fig 6j: Please clarify how this was calculated. My impression from Fig 6i is that there is not much difference between WT and 5xFAD in terms of amplitude/AUC.

I found the terminology for the different groups confusing at times, they use 'AD' and 'WT' but also distinguish between 'control' and 'experimental' groups. For the AD model they use 5xFADFezf2-CreER or ADFezf2-CreER. It is also not always clear what the wild-type is. Please clarify.

Throughout the text there are many typos, for example fig 1h, 'object interaction' instead of 'object interaction' or in Methods 'anaesthetised' instead of 'anesthetised'; the text would also benefit from English language editing. The figure legends need revision to improve understanding of the panels.

Reviewer #1 (Remarks to the Author):

The manuscript by Sun et al explores the role of cholinergic modulation of layer 5 mPFC neurons in novelty encoding in a mouse model of Alzheimer's disease. The manuscript itself showcases a number of powerful techniques that reveal some interesting findings with respect to the connectivity of layer 5 neurons and functional specificity of mPFC terminals in the supramammillary nucleus. Enthusiasm for the work is mixed, however. Important considerations of the animal model, and the neurobiological impact of the chemogenetic and optogenetic effects that were used need further discussion and or validation (more below). There are also several points that need to be addressed with respect to the experimental design and statistical analysis that limit my ability to evaluate of the robustness of the reported results. General and specific points listed below.

General comments

1. The manuscript focuses on the role of a subset of neurons in the mPFC and novel object recognition, although previous work has repeatedly shown that full lesions of the mPFC have no effect on novel object recognition (Ennacuer et al, 1997; Barker et al., 2007; Cross et al., 2013). This raises the question of circuit specificity and neurobiological relevance of the techniques used to manipulate the PFC as described in the study (more below).

Response: Thanks for your comments. As you mentioned, some studies with lesion showed that the mPFC is not essential for object recognition while others with drug inhibition showed the significant role of mPFC. Moreover, the lesion studies and the optogenetic inhibition results sometimes are inconsistent with each other. We added more discussion about this in discussion part from lines 622-637.

“It is also noteworthy that the selective excitotoxic lesion of mPFC indicated that the mPFC is not essential for object recognition⁶⁶⁻⁶⁸. However, other studies using drug inhibition of mPFC showed that infusion of glutamate antagonist into the mPFC before the test phase of the ORT can significantly impair the object recognition performance during the test phase^{64,69}. This reflected that the time window of manipulation on the mPFC is critical. In lesion studies, the structure lesion was occurred before the object recognition test and the test was done weeks or even months after the lesion surgery. The time window between the surgery and the behavior test may allow the brain to adapt the new neural network to process the novel object information during the behavior test. However, our optogenetic inhibition can control the neuron activity at milliseconds scale, which allows us more precisely to manipulate the activity of specific neural circuits in different phase of the task to understand their potential functions. Furthermore, in monkeys, the prefrontal neurons have been reported to show increases in responses to familiar stimuli compared to novel stimuli⁷⁰, which is consistent with our findings in rodent PFC neurons. These results suggested an involvement for the

medial prefrontal cortex in single-item recognition memory.”

2. With respect to the 5×FAD mice in particular, previous data has suggested that by 6 months of age significant plaque accumulation and signs of neurotoxicity are observed in both the hippocampus and layer 5 of the frontal cortex (Oakley et al. 2006; Eimer et al. 2013) Given the very clear role of the hippocampus in novel object recognition, especially in ORT with a short consolidation period as in the present study, as well as the role of SUM projections to DG in memory retrieval (Li et al., 2020), it is important to understand what PFC dysfunction is doing that is unique from the behavioral effects of hippocampal dysfunction. Sex as a biological variable should also be addressed, particularly given that the study was only run in female mice which appear to have greater amyloid deposition in this model. It would also be nice to know how consistent A β accumulation was between mice, and whether or not that correlated with any behavioral effects.

Response: The role of hippocampus on object recognition memory remains highly debated due to conflicting findings across temporary and permanent hippocampal lesion studies. Nevertheless, there are reports indicated that inactivation of dorsal hippocampus impaired the object recognition memory formation, consolidation and expression. While in our study, we found that inhibition of ET neurons in the mPFC only impaired the memory expression but not formation and consolidation. Therefore, it is likely that the object recognition memory is encoded and consolidated in dorsal hippocampus and other brain areas. Then the memory information transmitted to downstream brain areas include mPFC for retrieval. We have added this part in the Discussion part in lines 602-621: The hippocampus is another brain region responsible for memory formation, consolidation and expression. The role of hippocampus on object recognition memory remains highly debated due to conflicting findings across temporary and permanent hippocampal lesion studies⁵⁸. The inconsistent results may result from the different time window of manipulation, different brain regions of manipulation (dorsal hippocampus vs ventral hippocampus) and different delay time of the task. For example, studies have indicated that ventral hippocampus is essential for spatial memory^{56,59} but not for object recognition test^{41,56}. The inactivation of dorsal hippocampus impaired the novel object memory retrieval only in the tasks that contained longer delay time (24h) rather than short delay time (5min)^{60,61}. Nevertheless, there are other studies indicated that the dorsal hippocampus is also essential for object recognition memory with short delay time. Inactivation of dorsal hippocampus impaired object recognition memory formation, consolidation and expression^{62,63}. Our study together with previous studies found that mPFC is essential for object recognition memory expression⁶⁴ (Supplementary Fig. 7). In our results, we found that inhibition of the ET neurons in the mPFC only impaired object recognition memory expression but not formation and consolidation (Supplementary Fig. 7 and 8). Therefore, it is likely that the object recognition memory is encoded and consolidated in dorsal hippocampus and other brain areas. Then the memory information

transmitted to downstream brain areas include mPFC for retrieval⁶⁵.

There are previous studies showing that in 5×FAD mouse model, the female animals carried more A β burden and suffered from more severe behavior defects. Therefore, we chose female animals in our study. We have added this claim in the manuscript from lines 100-105.

We also added Supplementary Fig. 2f and 2g to explore how consistent A β accumulation was between mice, and whether or not that correlated with any behavioral effects. The data show a trend that more A β accumulation may lead to worse behavior performance. However, the A β accumulation and the behavior performance does not show strong correlation, indicating that there may be other factors that affect the animal's behavior. The description of the results can be found in lines 115-124.

3. The manuscript would benefit greatly from acknowledging more of the literature on this circuitry that has accumulated over the decades. In particular, I would suggest the authors take time to explore work by Martin Sarter, Mike Hasselmo, Huib Mansvelder, and Evelyn Lambe for additional information on what is already known about cholinergic modulation of deep layers of the mPFC, how this relates to cognitive function, and the impact of chronically increasing or decreasing ACh release in the mPFC with DREADDS relative to how the system is normally engaged.

Response: Thanks for the suggestions. We have added more discussion and cited more literatures to address the relation between our results and the known knowledge of the basal forebrain-mPFC circuits, which can be found in lines 674-682: Cholinergic modulation of cortical functions has been extensively studied. The cholinergic inputs to the mPFC have been indicated to be involved in attentional tasks⁷⁴⁻⁷⁶. The excitation of cholinergic receptors in the mPFC can enhance the sensory information input to optimize attention performance⁷⁴. Decreased acetylcholine release in the mPFC severely disrupted attention performance⁷⁷. Acetylcholine deficiency and cholinergic neuron degeneration have been observed in AD brains for decades¹⁹. Cholinergic fiber lesions have been found in many limbic brain areas, including the hippocampus and the mPFC³⁰. Specifically, the cholinergic input defects to layer VI of the mPFC have been identified in an AD mouse model⁷⁸.

Specific strengths

1. The power of the mPFC terminal manipulation in the SUM is really quite compelling. Again, understanding if this effect was unique to mPFCSUM interactions, or, if it reflected an indirect on SUMHippocampus modulation of memory would be very interesting.

Response: We thank the positive comments from the reviewer. In our results, we found that only manipulation of PFC-SUM pathway but not PFC-SI nor PFC-VTA affected the object recognition performance of the animals (Fig.3 and Supplementary Fig. 10). Whether this effect is also involved in the hippocampus need to be further investigated in the future study. We have discussed this pathway in

our Discussion part, which can be found in lines 589-601: Our results showed that the direct projection from mPFC to SUM could regulate the object recognition memory. These results revealed a novel pathway of how object recognition memory was processed in the brain, which could shed light on the understanding of neural circuitries responsible for object recognition memory and could help researchers understand how brain process, restore, and express object recognition memory. Further investigation is still needed to explore how the information delivered from the mPFC was processed within the SUM and transmitted to the downstream brain regions. In our results, we found that the ET neurons in the mPFC displayed stronger responses to familiar objects (Fig. 8). This similar response pattern was also observed in the vgat positive neurons in the SUM⁵⁶. Thus, the connectivity between the ET neurons in the mPFC and neurons in the SUM could potentially play the feedback circuit for object memory transmission back to hippocampus.

2. Throughout, the anatomical description and data are very interesting and well done.

Response: We thank the positive comments from the reviewer.

Issues remaining

1. Throughout the manuscript, it needs to be made clear what statistical comparisons were made. The F/T values, DF, and exact p-values should be included. It is also not clear that ANOVAs or T-tests were necessarily the best test of differences in central tendency for these groups. For example for photometry data, were signals mouse or trial weighted? There is a plot of a test on AUC, but what about event amplitude, event frequency. Why were these tests selected? Why these time windows? Given the effects of optogenetic stimulation on locomotor output in several conditions, a regression controlling for the effect of locomotor output on target-associated Ca²⁺ dynamics would be informative.

Response: We revised the statistics sheet and the figure legend to clarify all the analysis. For the fiber photometry data, the calcium signals were first averaged across trials from individual animals. The average plots were compared across different animals. We used AUC to characterize the amplitude of the calcium signal. We also provided the peak df/f to characterize the amplitude of the calcium signal in Supplementary Fig. 4, which is consistent with our AUC analysis. We also provided the number of calcium transient from different animals in Supplementary Fig. 4. There are no differences between Fezf2-CreER group and AD_{Fezf2-CreER} group. The analysis time window was selected according to the similar previous studies, which can be found in the Methods part from lines 380-385. We also provided the regression controlling for the effect of locomotor output on target-associated Ca²⁺ dynamics in Supplementary Fig. 4. Our results showed that the neither the amplitude of the calcium signal nor the number of calcium transient are correlated with the velocity of the animals during the test. The relevant results and description can be found in Supplementary Fig. 4 and lines 143-153.

2. Why was a 10 minute delay used for the ORT experiments? Were longer delays tested?

Response: In this study, we chose the 10 min delay according to the previous literature³⁸. We found that with 10 min of delay time, the animals in AD group and wild type group already showed different behavior performance (Supplementary Fig. 2). Therefore, we chose the 10 min delay time. We added more data in Supplementary Fig. 2e to test the effect of longer delay time in object recognition test. With longer delay time, the animals showed worse performance. However, the difference between 10 min delay and longer time delay is not significant. The description of the results can be found in lines 115-118.

3. Did the AD-Fezf2-CreER cross show a quantitatively similar pattern of Cre expression in the mPFC?

Response: We added Supplementary Fig. 3 to explore whether crossing between 5×FAD and Fezf2-CreER has any effects on the Cre expression in the mPFC. According to the FISH results, the crossing did not affect the number of Cre positive cells in the mPFC. The description of the results can be found in lines 128-135.

4. With respect to the photometry data in Figure 1, more information is needed. Later in the manuscript it is reported that Layer V neurons exhibit a greater response to familiar objects relative to novel. Is the reduced amplitude of evoked mPFC responses in the AD mice specific to the familiar object? Is activity overall reduced? Did response magnitude to the familiar object correlate with overall performance in this cohort?

Response: We are sorry that we didn't make it clear. In Fig. 1, we didn't separately analyze the response to familiar and novel objects. Instead we analyzed the overall response to the object interaction. We have revised the description in the results part, which can be found in line 136. The reduced amplitude of evoked mPFC responses in the AD mice is specific to the familiar object. The response magnitude to the familiar object is also correlated with the object recognition performance, which can be found in the Supplementary Fig. 19b and in lines 508-510.

5. For the c-fos/GFP examination depicted in Figure 1 M-P, what exactly was the N for each of the groups? Was there a control condition for the AD mice as well? Although the author's state there is "no obvious difference" in the number of GFP cells between AD and non-AD mice, the representative images suggest there are indeed fewer cells in the AD mice. It would be important to rule this out with a larger cohort and fewer neuron number could easily account for the difference in the photometry signal with no need to invoke the cholinergic system.

Response: The n value represents the number of animals we used. We also added the AD control group

in the Supplementary Fig. 5. As for Fig. 1, we apologized that we did not choose the best picture for presentation. We have replaced the pictures to avoid any misunderstanding. The data in Fig. 1n, Supplementary Fig. 3 and 6 showed that the neurons that expressed GCaMPs, Cre expression and the number of neurons in the mPFC showed no difference at 6 months of age between AD group and wild type control group.

6. For the data described in Supplemental Figure 3, was the total cell count (not just fos+) reduced in the AD model? In other words, did the PFC not recruit the structure, or, was there less of the structure to recruit?

Response: Thanks for the suggestion. We have added the Supplementary Fig. 6a-l to explore the neuronal loss in the mPFC, SI, SUM and VTA. There is no neuronal loss in these brain areas at 6 months of age in the AD mouse model. The relevant description can be found in lines 179-186.

7. For the optogenetics experiment in Figure 2, as well as the rest of them (e.g Figure 3, 7, Supplemental 10), how was specificity of opsin expression confirmed? How were stimulation parameters determined? When exactly is the laser being turned on? Why at that time? If augmented mPFC responses to familiar stimuli are what is important for ORT, were experiments with opto used to test this hypothesis directly? In some places it is noted that optical stimulation/inhibition affected locomotor output, but this is not addressed for every opto experiment and really should be to facilitate interpretation of the effects, particularly the impaired ORT in WT mice following optogenetic stimulation and inhibition.

Response: The specificity of the Fezf2-CreER animals were confirmed by checking the virus expression patterns. We found that most of the virus infected neurons are located in deep layers of the mPFC (Supplementary Fig. 1), which is consistent with the report that the Fezf2 mainly expressed in the deep layers of the cortex. The virus specificity was confirmed by immunostaining and the colocalization with the endogenous fluorescent signals (Supplementary Fig. 16 and 18). The virus we used in the optogenetics experiments for Fig. 2, 3, 7 and Supplementary figures are from the same batch. Therefore, the animals and the virus we used are highly specific.

To explore the role of Fezf2+ neurons or the cholinergic neurons in the different phase of object recognition test, the laser was delivered at different phase of object recognition test for 10 min to cover the whole duration of that phase. For Supplementary Fig. 20, the light was delivered when the center point of the animal entered the zone where the objects are located. The stimulation parameters were determined according to the previous studies as well as the behavior of the animals. For example, we found that high frequency (20 Hz) activation of Fezf2+ neurons in the mPFC can induce seizure. Therefore, we used low frequency (10 Hz) stimulation. We have revised the description of the

optogenetic experiments in the methods part, which can be found in lines 320-326.

We added the locomotion effect of manipulation of Fezf2+ neurons in the mPFC of WT group in Supplementary Fig. 7 and 8. We found that activation of Fezf2+ neurons in WT animals also increased locomotion, while inhibition of them has no effect on locomotion. We also added the discussion about the locomotion effect of the optogenetic experiments, which can be found in lines 655-673: In our optogenetic manipulation experiments, we found that activation of ET neurons in the mPFC increased the locomotion of the animals (Fig. 2 and Supplementary Fig. 8). However, our photometry results showed that the locomotion of the animals was not correlated with the neuronal activities of the ET neurons in the mPFC (Supplementary Fig. 4). Moreover, we found that inhibition of ET neurons in the mPFC had no effect on the animals (Supplementary Fig. 7). Therefore, we assume that the increase of locomotion is most likely to be caused by the unnatural optogenetic activation of the somas in the PFC. However, we found that activation of ET neurons in the AD(AD_{Fezf2-CreER}) group and wild type (Fezf2-CreER) group had similar effect on locomotion but opposite effect on object recognition (Fig. 2 and Supplementary Fig. 8). The neuronal activity of ET neurons in the mPFC showed no correlation with the locomotion of the animals (Supplementary Fig. 4). Therefore, the cognition alteration of the animals is unlikely to be related to the locomotion change. In our axon terminal manipulation experiments, we did find that inhibition of mPFC-VTA pathway decreased the locomotion of the animals. The mPFC-VTA pathway has been implicated to be involved in the social reward process and can be impaired by chronic stress⁷³, therefore, the locomotion effect of inhibition of mPFC-VTA pathway is more likely due to the emotional state change rather than movement disability. The exact functions of the mPFC-VTA pathway needs to be further investigated.

8. For Figure 5, seems surprising and potentially important if indeed there is no input from nucleus basalis (NBm) to layer 5 PFC. Some quantification here would be nice. The GFP-based anterograde tracing would also be strengthened by a quantification of AChE fiber density in PFC.

Response: We quantified the rabies tracing results mainly based on the Allen brain atlas. However, in the Allen brain atlas, there is no nucleus basalis (NBm). This brain area was named as magnocellular nucleus (MA), which is beside the posterior HDB. The rabies virus labeling in this brain area is not consistent across brain samples. This is probably due to the low efficiency of the RV labeling. Therefore, we didn't separate the NBm and HDB during our quantification.

We added the chat immunostaining results in Fig. 5i and 5j to show the cholinergic axonopathy progressions in the mPFC. The description of the results can be found in lines 378-381.

9. For the DREADD experiments, a CNO control group is necessary to control for the effects of a week of being exposed to low dose, back converted into clozapine. Also, important to confirm that in animals

that exhibited behavioral effects, virus expression was indeed confined to the horizontal band and did not spread to nearby cholinergic cell groups that project to other potentially important brain regions.

Response: We added the characterization of virus injection site in the HDB to confirm accuracy of the virus injection in Supplementary Fig. 18. We also added Supplementary Fig. 18I to explore the effect of chronic administration of CNO alone on animal behavior. Our results showed that the injection sites were mainly confined to HDB. The administration of CNO alone has no effect on the behavior performance of the animals. The description of the results can be found in lines 491-498.

Reviewer #2 (Remarks to the Author):

In their article, Sun and colleagues provide evidence that mPFC circuit impairment underlies decreased object recognition memory in a mouse model of Alzheimer's disease. The authors use state-of-the-art opto- and chemogenetic tools combined with in vivo behaviour to trace down an impaired neurocircuit in the 5×FAD mouse model. The paper thereby presents a novel brain circuit (ACh neurons BF  ET neurons in mPFC cortex  Undefined neurons in SUM) that is engaged in healthy subjects during Object Recognition, and is impaired in AD.

Overall, the study is carefully conducted with a high quality of the used methods and concisely described in appealing figure designs. Convincingly, the authors trace and describe a neurocircuit underlying object recognition memory in mice and pinpoint degeneration of cholinergic neurons as a hallmark in circuit dysfunction associated with memory loss. While it is not new that cholinergic neurons undergo degeneration in humans and various mouse models of Alzheimer's disease, the present work elegantly depicts the consequences on neurocircuits. (Over)activation of the remaining cholinergic system, or activation of second/third order neuronal populations however is sufficient to restore object recognition memory in the used AD model. Importantly, the authors transfer the progress of the last decade in systems neuroscience methods to a disease relevant model and highlight the opportunities using these methods outside of basic research to analyze disease progression and clinically relevant disease model in more detail.

Major:

1. Mouse model. The authors choose the 5×FAD mouse as a model system to analyze functional impairment of basal forebrain cholinergic – mPFC circuits to underly cognitive deficits in Alzheimer's disease. However, neither in the introduction nor in the discussion, the authors clarify why exactly this AD model has been chosen. Given the variety of different AD mouse models, the authors should clearly describe the rationale behind their decision and describe advantageous and possible disadvantageous of the model. The rationale should already be given in the introduction while the rest should be included in the discussion in more detail also comparing their results with other models and integrate into the literature.

Response: We have added the reason why we chose 5×FAD mouse model in the introduction part, which can be found in lines 62-66. We also discussed about the potential disadvantage of this animal model in the discussion part, which can be found in lines 729-737.

2. Development of cognitive dysfunction and disease progression. The authors provide convincing behavioral data at two different time points (2 and 6 month) upon which they decide to further perform their experiments in the older animals. While these experiments are a well-chosen basis of the whole

project, enthusiasm could be further increased if the authors would provide a more precise timeline of disease progression. While performing experiments again at multiple timepoints certainly is beyond the scope of reasonable request, a detailed immunohistochemical analysis in combination with further retrograde or anterograde tracing could provide important information. More precisely:

No behavioral phenotype is detected in 2-month old animals. When is the first decline in PFC-related memory impairment detectable? The 5×FAD mouse model shows plaques and gliosis already at the age of 2 month. For future studies in the same and other groups this would provide important information when to further expand on the interesting and important findings reported in the present manuscript

a) Does the memory decline coincide with cholinergic neuron loss in the basal forebrain and/or fiber loss in the mPFC?

b) Does fiber loss in the mPFC precede neuronal loss in the basal forebrain?

To address these points, we suggest the authors add in a two month time point with methods utilized in figure 5 (a-c), and provide a quantification of axonal integrity (d-g) for the already acquired 6 month time point and a 2 month timepoint. The nicely designed experiment indicates terminal loss precedes soma loss, as discussed by the authors. However we feel that given the presented variability in soma counts across individual mice at 6 months, inclusion of the 2-month time point and quantification of axonal integrity at 2 months and 6 months would greatly benefit the community and the impact of the presented work.

Response: Thank you for the suggestions. We have added Supplementary Fig. 12 and 15 to show the whole brain input to ET neurons in the mPFC and cholinergic inputs from basal forebrain of both *Fezf2-CreER* and *AD_{Fezf2-CreER}* animals. Our results showed that at two months of age, the whole brain inputs and the cholinergic input from the basal forebrain to the ET neurons showed no difference between AD group and wild type group. We also added Fig. 5i and 5j to show the cholinergic axonopathy progressions in the mPFC. With the development of AD, the cholinergic axonopathy increased dramatically. This result indicated that the cholinergic fibers keep on degenerating during the AD development. The description of the results can be found in lines 361-363 and 378-381.

In addition, our results showed that the 5×FAD mice at 6 months of age, but not at 2 months of age, spent less time with novel object compared to C57BL/6J mice (Supplementary Fig. 2c-e), which indicated that 6-month-old 5×FAD mice exhibited impaired object recognition memory. The behavior performance is consistent with the cholinergic axonopathy progressions in the mPFC.

3. The title of the paper implies that circuit impairment resides at the site of projection neurons of the mPFC, however, the authors elegantly identify basal forebrain cholinergic neuron soma and fiber loss as the main driver of circuit dysfunction. Hence the title of the paper should be slightly adjusted accordingly, in order to better disseminate message/target relevant audience.

Response: We have revised our title to “Acetylcholine Deficiency Disrupts Extratelencephalic Projecting Neuron Functions in the Prefrontal Cortex of Mouse Model of Alzheimer's disease” to include the information of the cholinergic neuron.

Minor:

1. Please clarify whether heterozygous/homozygous 5×FAD mice were employed in the study. If mixed groups were used, can the authors find any significant differences in ORT performance as a function of zygosity? Can this factor help explain phenotypic variance (ie, spread of data)

Response: We have clarified that the heterozygous 5×FAD mice were employed in the study, which can be found in the methods part. We didn't test the effect of zygosity. However, we did find that the ORT performance was correlated with the ET neuron response in the mPFC to the familiar object (Supplementary Fig. 19) and there is a trend that the more A β burden in the mPFC, the worse performance the animals exhibited (Supplementary Fig. 2f and 2g), although the R square did not reach significant difference to zero. Therefore, we think the ORT performance of the animals may be affected by the neuron activity in the mPFC, the A β burden and other factors.

2. For clarity, please refer to training/delay/testing with consistent phrases in main text and all other sections, (sometimes delay is referred to as ‘consolidation’, sometimes testing referred to as ‘expression phase’)

Response: We have revised the manuscript according to reviewer's suggestions.

3. Were differences in soma counts found in the rabies retrograde tracing experiments in Fez-cre mice? Statement in text is ambiguous and not further explained ‘We found that the proportion in most of the input brain areas were similar between 5×FAD and WT mice’. What precise presynaptic areas were affected? Was this limited to the basal forebrain as further described? What was the mean number of soma counted? Plots currently report proportion of total (%)

Response: We have plotted the total number of input cells in Supplementary Fig. 12 and 13. The total number and the proportion of the input brain areas we quantified showed no differences. Only the cholinergic inputs from the HDB were altered at 6 months of age, which was further described in Fig. 5.

4. Further typos in text should be revised. Also typos in axis label across should be checked for (for example figure1e, fig1k)

Response: We have revised the manuscript according to reviewer's suggestions.

Reviewer #3 (Remarks to the Author):

This is an interesting study investigating the neural circuitry underlying impaired novel object recognition in Abeta plaque-bearing 5×FAD mice. The study provides novel mechanistic insight into the loss of novel object preference in this mouse model by showing that decreased cholinergic input to ET neurons in the mPFC appears to be the major driver of this deficit. However, several technical, analytical and conceptual issues reduce my enthusiasm for the study, and the relevance of their findings for the human condition remain uncertain.

The study is motivated by 'clinical and animal model studies which demonstrated impairment of object recognition memory in Alzheimer's disease (AD)' and seeks to examine 'the underlying neural circuit mechanism'. However, to achieve this the authors need to be much more precise (e.g, by using adequate references) at which stage of the disease such deficit occurs, and this then should inform the choice of mouse model. If such deficits occur in individuals with AD dementia then using an Abeta model alone may not be sufficient to model this stage, and what we learn from the mice may not be relevant to human disease. Alternatively, they could remove all claims of human relevance and simply state that they are investigating the circuitry that underlies novel object recognition memory deficit in this particular mouse model (also keeping in mind that other models may not show deficits or deficits before plaques emerge, and the mechanisms may be different).

Response: Thanks for the suggestions. We have cited two more literatures to show that in the AD patients, some of them also can show object recognition defects^{3,4}, which can be found in line 38.

Information on statistical analysis is inadequate in my opinion, and as far as I can ascertain the relevant information is not present in the results section, figure legends or methods which makes it impossible to assess the validity of some of the results. They list statistical comparisons they made in one of the excel files (303849_0_source_data_891366) but it doesn't contain all the info they claim to have provided in the reporting table (eg, normality testing)?

Response: We revised our statistics sheet to clarify all the statistical analysis we used in the manuscript.

5×FAD are available both on mixed background and congenic BL/6J background and it is important to know which they used for their cross-breedings as they may end up comparing mice on different background strains which is a confound. Can they specifically state the backgrounds of their mice and that all genotypes used are on the same background? Also, it appears that they are only using female mice, can they please clarify and also provide the rationale for this?

Response: All the animals we used in our experiments are in C57BL/6J background. We have revised our methods part to clarify the genetic background. We also cited more literatures to show that the

female 5×FAD mice usually carried more A β burden and suffered from more severe behavior defects, which can be found in lines 102-105: Specifically, there are reports showing that 5×FAD mice exhibit sex biased pathology progression and female animals showed more severe symptoms and A β accumulation^{35,36}. Therefore, we chose female 5×FAD animals in our following experiments.

The focus is on layer 5 mPFC in 5×FAD mice but I would like to see as a negative control and to demonstrate specificity that the same deficits are not present in another brain region, in particular hippocampus which is thought to be involved in novel object recognition.

Response: We added Supplementary Fig. 9 to explore whether activation of brain areas other than mPFC of 5×FAD animals can rescue the object recognition impairment. With the ventral hippocampus as an example, our results showed that activation of the ventral hippocampus of the 5×FAD mice can not rescue the impaired ORT performance. The description can be found in lines 235-245.

Fig 1, calcium data: the data suggest that the amplitude of population calcium transients in 5×FAD is reduced compared to WT. Amplitude difference may also be explained by fewer cells expressing GCaMP in 5×FAD and other technical confounds, which needs to be clarified.

Response: The data in Fig. 1n, Supplementary Fig. 3 and 6 showed that the total neuron and neurons expressed Gcamp and Cre in the mPFC showed no difference between AD and control group. Therefore, we think the reduced calcium amplitude is due to the animals rather than the Gcamp expression.

Fig 1, c-fos data: There is almost no background in 'AD NOR' compared to 'WT NOR'. C-fos antibody can also stain plaques. My first thought was that given this, the higher background in the WT-NOR could be just because more excess antibody was sequestered by the plaques in the 5×FAD group. But actually, the dark cross in the WT-NOR suggests over-exposure and tiling effects, and the GcaMP in that group also looks odd compared to the others. C-fos quantification can be especially sensitive to brightness/background (and may be masked by the plaques), so I think this is an issue that needs clarification as it may impact on their results.

Response: We apologized that we did not make it clear. The different background in the Supplementary Fig. 5 is because the images were captured by different microscope. The WT NOR group and the AD control group were imaged with Leica sp8 confocal microscopy; the WT control group and the AD NOR group were imaged with Olympus VS120 slider scanner. We have clarified the imaging methods in the methods part, which can be found in lines 204-206.

Figs 2d, e: Is this the same mouse shown here? Please clarify.

Response: Yes.

Fig 2g: It appears that the negative control for Chr2 in ADFezf2-CreER is expression of GFP but I would also like to see the effects of Chr2 expression in Fezf2-CreER.

Response: Our results showed that activation of Fezf2 positive neurons in the mPFC of Fezf2-CreER animals during the test phase impaired the object recognition. The results can be found in Supplementary Fig. 8.

Fig 6i: 'on multiple animals', please clarify number of animals used. This is a general issue throughout that information on number of animals and statistical tests is missing.

Response: We have added all the number of animals we used in the figure legends.

Fig 6j: Please clarify how this was calculated. My impression from Fig 6i is that there is not much difference between WT and 5×FAD in terms of amplitude/AUC.

Response: First we calculated the average plots across different trials of individual animals with MATLAB, which were shown in Fig. 6f and 6h. The Fig. 6i is the average plot of different animals. The AUC of the average plots of different animals from-2-4s were calculated with GraphPad Prism version 6.0. The 6j plots the AUC of different animals we tested. Two-tailed unpaired t test was used for comparison. We have revised our methods part to make it clear, which can be found in lines 400-404.

I found the terminology for the different groups confusing at times, they use 'AD' and 'WT' but also distinguish between 'control' and 'experimental' groups. For the AD model they 5×FADFezf2-CreER or ADFezf2-CreER. It is also not always clear what the wild-type is. Please clarify.

Response: We have revised the manuscript according to reviewer's suggestions.

Throughout the text there are many typos, for example fig 1h, 'object interaction' instead of 'object interaction" or in Methods 'anaesthetised' instead of 'anethetised'; the text would also benefit from English language editing. The figure legends need revision to improve understanding of the panels.

Response: We have revised the manuscript according to reviewer's suggestions.

Reviewers' Comments:

Reviewer #1:

Remarks to the Author:

The authors were incredibly responsive to the previous round of review, resulting in an absolutely thorough examination of this pathway and relevance to object recognition. My suggestions at this point are geared towards enhancing the ease of interpretation.

1. For the statistics, please do reference the results of normality tests to confirm the appropriateness of the reported results. I cannot find this information.

2. With the sheer amount of data now in the manuscript, it becomes quite difficult to compare all the different mouse lines employed. The cornerstone of this work is the structure-function relationships, so for the 5×FAD cross lines, please confirm that the behavior of the cross (e.g. AD-Chat or AD-Fezf2-CreER) is the same as the 5XFAD, and different from a WT or litter-matched control in ORT. For example, in Supp. Fig 4 there does not appear to be a difference in interaction time between the WT and AD-Fezf2-CreER. How then do we relate the change in neural activity in these animals to the behavior of the 5XFAD cross? Apologies if this information is contained in the manuscript, but I could not find it.

3. As fiber placement and transfection will have a major impact on light output measured with photometry, a clear indication of the location of recordings for each major comparison between experimental groups would be very informative. Something similar to what is in Suppl. Fig. 18.

Reviewer #2:

Remarks to the Author:

Overall the authors addressed the points raised.

There is just a small point in response to 'major point 2'

Some arrows seem to have jumped in your supplementary figure 15 top panel. Not sure about the reported percentage. Do the authors mean number of:(Gfp+ and Chat+)/ Gfp+?

Please clarify

Reviewer #3:

Remarks to the Author:

I'd like to thank the authors for taking the time to address my comments.

Unfortunately, some of my comments are dealt with only superficially – for example, they continue to refer to 'AD patients' and refer to literature about individuals with dementia (due to probable AD) however the mouse model they are using in this study, if anything, reflects early preclinical AD. This means conclusions from this mouse may or may not be relevant to human AD, and if they cannot provide evidence for impaired object recognition in individuals with preclinical AD (eg, asymptomatic familial cases) they should refrain both in abstract and main text from linking their findings in mice to humans.

Re statistics – I thank the authors for providing more detail on the statistical tests in the figure legends however justification for some of these tests is still missing as far as I can ascertain (e.g., normality testing for t-tests)?

Re choice of sex – the justification for using female mice because they have more pathology is not really convincing since they have not tested this in their specific cohort. Instead, they should mention this as a limitation of the study.

Re hippocampal experiments – the authors chose ventral hippocampus as a control region. I'd thought that based on available literature dorsal hippocampus would have been the better choice. please clarify.

Re calcium data in fig. 1 – my concern was that amplitude differences between WT and AD model are due to differences in the number of GCaMP-positive neurons. They refer to several figures in main text and supplementary material but none of these shows counts of cells that are GCaMP-positive so my concern has not really been addressed.

Re cfos data – in response to my concerns that images look different between groups which may impact the quantification the authors responded they used different microscopes. i don't find this reassuring and even more important to confirm the validity of the results.

Re fig 6j - does this fig show the AUC calculated from fig 6i?

Reviewer #1 (Remarks to the Author):

The authors were incredibly responsive to the previous round of review, resulting in an absolutely thorough examination of this pathway and relevance to object recognition. My suggestions at this point are geared towards enhancing the ease of interpretation.

1. For the statistics, please do reference the results of normality tests to confirm the appropriateness of the reported results. I cannot find this information.

Response: We have added the normality tests and updated the statistics table and figure legends.

2. With the sheer amount of data now in the manuscript, it becomes quite difficult to compare all the different mouse lines employed. The cornerstone of this work is the structure-function relationships, so for the 5×FAD cross lines, please confirm that the behavior of the cross (e.g. AD-Chat or AD-Fezf2-CreER) is the same as the 5XFAD, and different from a WT or litter-matched control in ORT. For example, in Supp. Fig 4 there does not appear to be a difference in interaction time between the WT and AD-Fezf2-CreER. How then do we relate the change in neural activity in these animals to the behavior of the 5XFAD cross? Apologies if this information is contained in the manuscript, but I could not find it.

Response: We appreciate the referee's comments. In AD_{Fezf2-CreER}, AD_{ChAT-Cre} and 5XFAD mouse line, the AD animals didn't show clear preference towards novel objects (Fig. 2, 7 and Sup Fig. 2) as in wild type animals (Sup Fig. 7, 8, 16, 17). The higher delta object recognition index represents the clearer preference towards novel objects. Recognition memory was scored using a recognition index for each mouse with a formula $(N \text{ or } F) / (N + F) \%$, while the delta recognition index = recognition index(N) - recognition index(F).

Sup Fig. 4d showed the total number of interaction times with both familiar and novel objects of AD animals and WT animals. The total number showed no difference, which indicated that the AD animals could explore the objects in the arena. However, they couldn't remember which object is novel and which is the old one. We have revised the legend of the Sup Fig. 4d to avoid misunderstanding.

3. As fiber placement and transfection will have a major impact on light output measured with photometry, a clear indication of the location of recordings for each major comparison between experimental groups would be very informative. Something similar to what is in Suppl. Fig. 18.

Response: We have added the placement of fiber photometry experiments in Fig. 6, Supplementary Fig. 4 and 18.

Reviewer #2 (Remarks to the Author):

Overall the authors addressed the points raised.

There is just a small point in response to ‘major point 2’

Some arrows seem to have jumped in your supplementary figure 15 top panel. Not sure about the reported percentage. Do the authors mean number of:(Gfp+ and Chat+)/ Gfp+?

Response: We thank the referee for the suggestions. We have revised the supplementary figure 15 to make sure the arrows are in the right position. We also changed the figure legend of supplementary figure 15b to (GFP+ and Chat+)/ GFP+.

Reviewer #3 (Remarks to the Author):

I'd like to thank the authors for taking the time to address my comments.

Unfortunately, some of my comments are dealt with only superficially – for example, they continue to refer to 'AD patients' and refer to literature about individuals with dementia (due to probable AD) however the mouse model they are using in this study, if anything, reflects early preclinical AD. This means conclusions from this mouse may or may not be relevant to human AD, and if they cannot provide evidence for impaired object recognition in individuals with preclinical AD (eg, asymptomatic familial cases) they should refrain both in abstract and main text from linking their findings in mice to humans.

Response: We thank the referee for the comment. We have added more literatures to show that impaired object recognition was found in asymptomatic familial AD cases especially for those who carried the presenilin-1 mutation, which was also carried by the mouse model we used in the present study.

We have revised the introduction part in lines 38-40: Specially, the object recognition memory is impaired in sporadic Alzheimer's disease, familial Alzheimer's disease, and in asymptomatic carriers of familial Alzheimer's disease with presenilin-1 mutation⁵⁻⁹.

The added literatures can be found below:

Mario A.Parra, S. D., Sharon Abrahams, Robert H.Logie, Luis Guillermo Méndez, Francisco Lopera. Specific deficit of colour–colour short-term memory binding in sporadic and familial Alzheimer's disease. *Neuropsychologia* 49, 1943-1952 (2011).

Tiedt, H. O., Benjamin, B., Niedeggen, M. & Lueschow, A. Phenotypic Variability in Autosomal Dominant Familial Alzheimer Disease due to the S170F Mutation of Presenilin-1. *Neurodegener Dis* 18, 57-68 (2018).

Parra, M. A. et al. Visual short-term memory binding deficits in familial Alzheimer's disease. *Brain* 133, 2702-2713 (2010).

Arango-Lasprilla, J. C., Cuetos, F., Valencia, C., Uribe, C. & Lopera, F. Cognitive changes in the preclinical phase of familial Alzheimer's disease. *J Clin Exp Neuropsychol* 29, 892-900 (2007).

Parra, M. A. et al. Brain Information Sharing During Visual Short-Term Memory Binding Yields a Memory Biomarker for Familial Alzheimer's Disease. *Curr Alzheimer Res* 14, 1335-1347 (2017).

Re statistics – I thank the authors for providing more detail on the statistical tests in the figure legends however justification for some of these tests is still missing as far as I can ascertain (e.g., normality testing for t-tests)?

Response: We have added the normality tests and updated the statistics table and figure legends.

Re choice of sex – the justification for using female mice because they have more pathology is not really convincing since they have not tested this in their specific cohort. Instead, they should mention this as a limitation of the study.

Response: We have added more discussion in lines 742-746.

Another limitation of the present study is that we only explored the structural and functional alteration in female AD animals. The structural and functional alteration of neural circuits in male AD animals and why the female AD animals showed more severe symptoms is also worth exploring in future studies.

Re hippocampal experiments – the authors chose ventral hippocampus as a control region. I'd thought that based on available literature dorsal hippocampus would have been the better choice. please clarify.

Response: We agree with the reviewer that it is interesting to explore the hippocampal circuit alteration in the AD mouse model and to find out which circuit defect can also lead to object recognition memory impairment. The role of hippocampus on object recognition memory remains highly debated due to conflicting findings across temporary and permanent hippocampal lesion studies⁶³. Since the hippocampal formation contains multiple brain areas. Even the dorsal hippocampus alone includes CA1, CA2, CA3 and dentate gyrus. Which brain area is involved in object recognition memory and which brain area is compromised in AD mouse model is not clear. The inconsistent results may result from the different time window of manipulation, different brain regions of manipulation (dorsal hippocampus vs ventral hippocampus) and different delay time of the task. It will take lots of work to figure it out, which is beyond the scope of the present study.

We have added more discussion about this topic in lines 624-627: There have been reports showed that altered hippocampal circuits in AD mouse model can lead to abnormal spatial memory and anxiety expression^{34,36,71,72}, whether and how alteration of hippocampal circuits caused by AD affect object recognition memory needs to be investigated in future studies.

Re calcium data in fig. 1 – my concern was that amplitude differences between WT and AD model are due to differences in the number of GCaMP-positive neurons. They refer to several figures in main text and supplementary material but none of these shows counts of cells that are GCaMP-positive so my concern has not really been addressed.

Response: We apologized that we mislabeled the legend of Fig. 1. The Fig. 1n represent the quantification of the number of GCaMP-positive neurons in both AD and WT group. The results showed that the number of GCaMP-positive neurons in the PFC of both groups showed no differences ($p=0.8157$, one-way ANOVA followed by Tukey's post hoc correction). We have revised the legends to avoid misunderstanding.

Re cfos data – in response to my concerns that images look different between groups which may impact the quantification the authors responded they used different microscopes. i don't find this reassuring and even more important to confirm the validity of the results.

Response: We apologized that the background in Sup Fig. 5 is different due to the acquisition of different microscopes. When we compared these figures together, we adjusted the contrast and the brightness of

the figure across different groups. As shown below, the adjustment of the background affects the display effect of the figure, but does not affect the statistical results.

Re fig 6j - does this fig show the AUC calculated from fig 6i?

Response: Yes. We have revised the legend of Fig. 6 to show that the Fig. 6j is the quantification of the AUC in 6i, which can be found in line 1232.

Reviewers' Comments:

Reviewer #1:

Remarks to the Author:

The authors have addressed my remaining concerns. Congratulations on an impressive set of data.

Reviewer #2:

Remarks to the Author:

We thank the authors for their revision of the text. In light of the novel circuit elements described in the manuscript using multiple complimentary methods, and their relation to AD phenotypes, we consider the manuscript as ready for publications in nature

Reviewer #3:

Remarks to the Author:

The paper remains very poorly written and should be carefully proof-read by a native speaker. In addition, the sloppy use of terminology and oversimplifications when comparing animal and human data remains for me a major concern.

For example:

Terms like 'AD mice' or 'AD animals' are misleading and should be avoided. Why not refer to the mice by the model name (ie, 5xFAD)?

They should also specify this in the abstract and not just write 'AD mouse model'

'Object recognition dementia was found in both AD patients and mouse models' – the term 'object recognition dementia' does not exist

In that context, they now cite additional studies that object recognition memory is impaired even in asymptomatic individuals with familial AD – this is not very convincing as the tests used are very variable between studies, and it remains unclear how they the object recognition task used in this study (in mice) translates to the human condition

'A series of studies have shown that cognition impairment correlates with the prefrontal cortex (PFC) dysfunction in both AD patients and mouse models – the term 'cognition impairment' is imprecise and certainly not all cognitive impairments in AD are due to PFC dysfunction?

in line 45-47 – they need to specify if humans or mice

in line 48-49 – 'in the mouse model of AD crossed with Thy1-GFP mice' – unclear which model they refer to

line 51-52: 'Functional studies indicate that ET neurons in the PFC play a key role in cognition' - the term cognition is very broad and they should be more precise

Unfortunately, there are many other examples throughout the text.

Reviewer #1 (Remarks to the Author):

The authors have addressed my remaining concerns. Congratulations on an impressive set of data.

Response: We thank the reviewer for raising important questions and helping us improve our manuscript.

Reviewer #2 (Remarks to the Author):

We thank the authors for their revision of the text. In light of the novel circuit elements described in the manuscript using multiple complimentary methods, and their relation to AD phenotypes, we consider the manuscript as ready for publications in nature

Response: We thank the reviewer for raising important questions and helping us improve our manuscript.

Reviewer #3 (Remarks to the Author):

The paper remains very poorly written and should be carefully proof-read by a native speaker. In addition, the sloppy use of terminology and oversimplifications when comparing animal and human data remains for me a major concern.

For example:

Terms like 'AD mice' or 'AD animals' are misleading and should be avoided. Why not refer to the mice by the model name (ie, 5xFAD)?

They should also specify this in the abstract and not just write 'AD mouse model'

Response: We have revised the main text to directly refer to the model name in the abstract and in the main text.

'Object recognition dementia was found in both AD patients and mouse models' – the term 'object recognition dementia' does not exist

Response: We have revised the object recognition dementia to object recognition memory impairment.

In that context, they now cite additional studies that object recognition memory is impaired even in asymptomatic individuals with familial AD – this is not very convincing as the tests used are very variable between studies, and it remains unclear how the object recognition task used in this study (in mice) translates to the human condition

Response: We assume that the reviewer's concern is that it remains unclear how the object recognition task used in this study (in mice) may translate to the human condition.

We cited human studies that mainly used visual paired comparisons task (VPC) to test the object recognition ability of the participants. According to the literatures, the VPC task in human studies is comparable with the one-trial novel object recognition (NOR) test and delayed non-matching to sample (DNMS) in rodents. We have added the literatures in the main text (lines 42-45).

Nevertheless, we agree with the reviewer that how our results in the present study translate to the human conditions is unclear since it is difficult to target specific neuron types in human brains. And we have

clearly stated in the discussion part that the results found in this study need to be further investigated in the future for human treatment.

'A series of studies have shown that cognition impairment correlates with the prefrontal cortex (PFC) dysfunction in both AD patients and mouse models – the term 'cognition impairment' is imprecise and certainly not all cognitive impairments in AD are due to PFC dysfunction?

Response: We have revised the term cognition in the main text to more precise descriptions such object recognition memory or short-term memory.

in line 45-47 – they need to specify if humans or mice

Response: We have added more details as requested by reviewer.

in line 48-49 – 'in the mouse model of AD crossed with Thy1-GFP mice' – unclear which model they refer to

Response: We have added the genotype of the AD mouse model in the main text as requested by reviewer.

line 51-52: 'Functional studies indicate that ET neurons in the PFC play a key role in cognition' - the term cognition is very broad and they should be more precise

Response: We have revised the term cognition in the main text to more precise descriptions such object recognition memory or short-term memory.

Unfortunately, there are many other examples throughout the text.

Response: We apologized about the English writing and the imprecise terminology use in the manuscript. We have carefully checked and revised the manuscript according to the suggestions. We thank again for the reviewer's patience and time for helping us improve our manuscript.